# Membrane lipid composition modulates the organization of VDAC1, a mitochondrial gatekeeper
Elodie Lafargue[1,4], Jean-Pierre Duneau[2,4], Nicolas Buzhinsky [1], Pamela Ornelas [3], Alexandre Ortega[1], Varun Ravishankar[2], James Sturgis [2], Ignacio Casuso [1] ✉ & Lucie Bergdoll [2] ✉

VDACs, the most abundant proteins in the outer mitochondrial membrane (MOM), are crucial for mitochondrial physiology. VDAC regulate metabolite and ion exchange, modulate calcium homeostasis, and play roles in numerous cellular events such as apoptosis, mitochondrial DNA (mtDNA) release, and different diseases. Mitochondrial function is closely tied to VDAC oligomerization, influencing key processes like mtDNA release and apoptosis, but the molecular drivers of this oligomerization remain unclear. In this study, we investigate the effects of three major MOM lipids on VDAC assemblies using atomic force microscopy and molecular dynamics simulations. Our results show that phosphatidylethanolamine and cholesterol regulate VDAC assembly, with the formation of stable lipid–protein organization of various size and compaction. Deviations from physiological lipid content disrupted native-like VDAC assemblies, highlighting the importance of lipid environment in VDAC organization. These findings underscore how lipid heterogeneity and changes in membranes influence VDAC function.

The Voltage-Dependent Anion channel (VDAC) is by far the most abundant protein (>50%)[1,2] in the Mitochondrial Outer Membrane (MOM), playing crucial roles in many cellular functions. VDAC is primarily known for facilitating the passage of ions and metabolites like ATP and ADP across the membrane, but is also involved in numerous mitochondrial regulatory processes such as apoptosis[3–5], calcium homeostasis[6,7], mtDNA release[8], and lipid scrambling[9]. VDACs act as gatekeepers to the mitochondria, anchoring different cytosolic proteins that interact with the organelle[10–12]. In mammals, VDAC exists as three isoforms—VDAC1, VDAC2, and VDAC3— each able to transport ions and metabolites, but contributing uniquely to mitochondrial function as they are involved in different pathways, interacting with different partners.

VDAC oligomeric organization in the membrane has been shown to critically influence several cellular events. This organization plays a central role in processes such as lipid scrambling[9], mitochondrial DNA release[8], and apoptosis[13], which are crucial for maintaining cellular homeostasis and responding to stress signals. Despite its significance, the molecular forces driving VDAC oligomerization remain poorly understood. Elucidating these forces is critical for understanding how VDAC oligomerization is regulated under physiological and pathological conditions, such as in

cancer, neurodegeneration, and metabolic diseases, where VDAC function is often disrupted.

Lipids play major roles in VDAC's function[14–16] and can regulate its interaction with several partner proteins[12,17]. MOM lipids can vary significantly under different physiological and pathological conditions, such as during apoptosis, diseases, and neurodegenerative disorders[18–20]. For instance, cancer cells often exhibit altered lipid metabolism[18], leading to changes in the composition of mitochondrial membranes, while in apoptosis, specific lipids like cardiolipin (CL) undergo remodeling to facilitate the release of apoptotic factors. In neurodegenerative diseases such as Alzheimer's and Parkinson's, mitochondrial lipid profiles are disrupted[19], with increased levels of cholesterol (Chol) and changes in phospholipid composition contributing to mitochondrial dysfunction and disease pathology[21,22]. VDAC is implicated in all these processes. Hence, understanding how these lipid variations impact VDAC oligomerization is essential for understanding VDAC function in various cellular processes, especially under stress conditions.

Atomic Force Microscopy (AFM, Fig. 1a-c) is well-suited for studying membrane protein organization at the molecular level. VDAC, a 19-strand β-barrel, exhibits a distinct 4.5 nm pore in AFM (Fig. 1d). In 2007, two independent AFM studies on native MOM from yeast[1] and potato tuber[2]

[1]Turing Centre for Living System, Aix-Marseille Université, U1325 INSERM, DyNaMo, Marseille, France. [2]Laboratoire d'Ingénierie des Systèmes Macromoléculaires, CNRS - Aix Marseille Université, Marseille, France. [3]Department of Structural Biology, Max Planck Institute of Biophysics, Frankfurt, Germany. [4]These authors contributed equally: Elodie Lafargue, Jean-Pierre Duneau. ✉e-mail: ignacio.casuso@inserm.fr; lbergdoll@imm.cnrs.fr

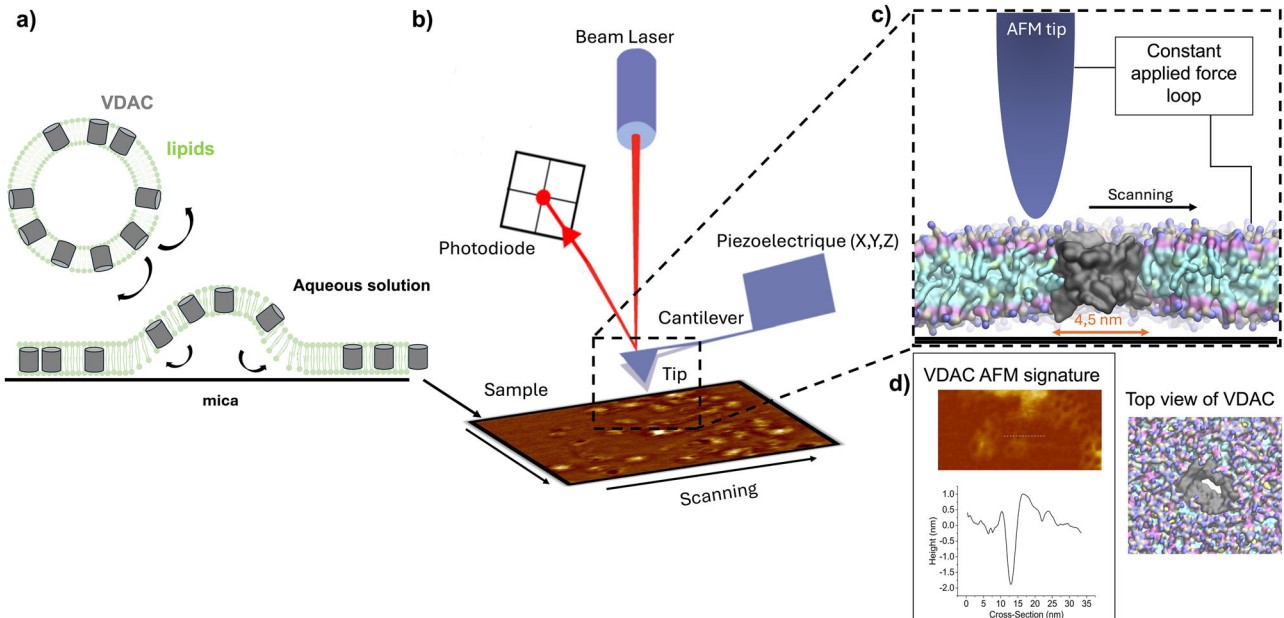

**Fig. 1 | General schematic of AFM experiment. a** VDAC1 reconstituted in vesicles of different lipid compositions was adsorbed on a mica surface. Following adsorption, the sample is thoroughly rinsed and transferred to the AFM system, leaving no free-floating particles in the solution within the imaging chamber. **b** Diagram of AFM principle. The tip, positioned on a lever, scans the sample (the tip oscillates). A laser beam is used to detect the bending of the lever during the scanning. The laser is reflected by the lever on the photodiode. In AFM imaging, the lateral (XY) resolution is determined by the size of the apex of the AFM tip (extremity of tip), typically ranging from 0.5 to 2 nm, which is sufficient for submolecular imaging. The vertical (Z) resolution is higher, achieving ~1 Å. **c** Zoom from b on the interaction between the tip and the sample surface. The sample-to-tip distance is controlled by the feedback loop to keep the force applied to the sample constant. **d** AFM signature of a VDAC pore and its profile showing a diameter similar to the expected size of 4.5 nm.

revealed multiple VDAC assemblies, predominantly large, densely packed proteins. While these studies are foundational, the presence of other proteins and uncontrolled lipid compositions prevent deciphering the effect of lipids on VDAC assemblies.

A previous study has demonstrated that CL and phosphatidylglycerol (PG), two lipids present in trace amounts in the MOM but key for mitochondrial metabolism and apoptosis, can influence VDAC oligomerization[23]. In this study, we investigated three major lipid components of the MOM: phosphatidylcholine (PC), phosphatidylethanolamine (PE), and Chol—a key MOM lipid known to interact with VDAC, and whose concentration can vary in the membrane under different physiological or pathological conditions[21,22,24,25]. While the interactions of VDAC1 with PE and Chol are established[16,26], their roles in VDAC1 organization and clustering remain poorly understood.

Using a combination of AFM and molecular dynamics (MD) simulations, we demonstrate that VDAC1's organization is strongly influenced by the membrane's lipid composition, with VDAC1 actively influencing its lipid environment to form specific assemblies. This lipid sensitivity enables VDAC1 to rapidly reorganize within the MOM in response to changing conditions.

## Results
To study VDAC1 assembly in an environment close to the MOM while controlling all parameters, we used purified VDAC1 reconstituted in mixtures of three of the main lipids of the MOM: two phospholipids (POPC and POPE) and Chol. VDAC1-containing liposomes were imaged by AFM (Fig. 1). Lipid to protein ratios 3:1 (w:w, corresponding to a 140:1 molar ratio) were selected as VDAC density similar to that observed in native MOM[1,2]. All experiments were performed in triplicate at temperatures between 33 and 35 °C to avoid lipidic phase separation (Fig. S1).

In the following sections, we use the term "structures" to refer to well-defined protein organizations detectable by wide-field AFM imaging. These include both aggregates, composed solely of proteins, and arrangements, which involve both protein-protein and lipid-mediated interactions. In contrast, "clusters" refer specifically to subsets within these structures where proteins are in direct contact, separated by no more than 5.6 nm.

### Lipid composition drastically impacts VDAC1 organization
Using POPC, POPE, and Chol, we reconstituted VDAC1 in various lipid environments. Supplementary Fig. 2 is a gallery of images showing drastically different shapes, compaction, and density of VDAC proteins, visible in lighter colors on a homogenous dark-brown lipid background. These observations, reproducible across experiments, highlight the diverse organizational patterns triggered by changes in lipid composition and hence the crucial role of lipids in regulating VDAC1 assemblies. In particular, dense assemblies of tightly packed proteins (Fig. S2f, g), in contrast to the more loosely arranged configurations (Fig. S2b, d), highlight the variability due to lipid types and ratios. In subsequent sections, we focus on these structures, their specific configurations under different lipid conditions, and their impact on VDAC1 assembly.

### VDAC1 forms protein-lipid structures: honeycombs
We reconstituted VDAC1 in two primary MOM lipids, POPC and POPE, in ratios closely resembling those of the native MOM (65:35 w:w, Table S1 describes the lipid ratios in the MOM under healthy conditions[27]). We observed the emergence of very stable (over several minutes) and densely packed assemblies composed of VDAC molecules and lipids, which we term "honeycombs," due to their resemblance to the hexagonal structure made by bees (Fig. 2a).

Appearing organized at first glance, a more detailed analysis based on a digitized model of the locations of the proteins identified by AFM (Fig. 2b, see Methods) reveals intrinsic disorder within these assemblies, where proteins are arranged in a noncrystalline manner, also termed glass-like organization. The term "glass-like" characterizes both disorganized crowding and amorphous assembly[28]. Our model of the VDAC honeycomb, generated using AFM positions of the VDAC molecules and MD

simulations (see Methods and below), reveals both direct and lipid-mediated protein–protein interactions. This underscores the critical role of lipids in honeycomb formation. The innermost lipid shell (yellow in Fig. 2b), represents lipids directly interacting with the protein (~20 per leaflet; Fig. S3). Notably, these honeycombs are very stable as the lipids act as "mortar" keeping the proteins near each other, but not always in direct contact. Interestingly, the honeycomb assemblies (formed within membranes containing only POPC and POPE) closely resembled the organizational patterns observed in a subset of the population within the native MOM of potato tuber[2] and yeast[1] previously visualized by AFM. Additionally, negative staining imaging of outer membrane vesicles derived from *Neurospora crassa* (a fungus, often used for VDAC and mitochondria stu-

dies) revealed a similar pattern of VDAC organization within the native membrane (Fig. 2c, d). Using only the two main lipids of the MOM is sufficient to recreate native VDAC organization in a reconstituted membrane.

### The Honeycombs are parallel assemblies of VDAC1

We developed a new method named "nanobead-mediated orientation determination," where a gold nanoparticle of 5 nm, functionalized with a Ni-NTA group, is attached to the 6His-tag of VDAC1 facing the intermembrane space (IMS) side of the protein (Fig. 3a). Using this method, we showed that VDAC is associated into parallel organization within the honeycombs (Fig. 3b, c), forming domains of parallel proteins that extend for tens of nanometers. We were able to image both orientations, the IMS-facing and the cytoplasm-facing assemblies (Fig. S4). During reconstitution, VDAC forms parallel honeycomb structures within vesicles, exhibiting both possible orientations (cytoplasmic or IMS side), subsequently adhering to mica for AFM imaging and revealing dual orientations. The parallel organization of VDAC, along with the presence of honeycombs in native MOMs, confirms the physiological relevance of this organization.

### Cholesterol modulates the compaction of the honeycomb assemblies

Chol is a major component of MOMs (between 5% and 10% w:w; see Table S1) and is known to interact directly with VDAC1 at several identified binding sites[27,29]. Although honeycombs can form in POPC/POPE membranes without Chol, we investigated the effect of cholesterol on VDAC1 organization by doing a titration from 2 to 10% (Fig. 4). Even at modest Chol levels, we observed significant alterations in the assemblies, leading to a wide range of behaviors and structures, as depicted in Fig. 4.

Remarkably, the addition of only 2% Chol to the POPC/POPE lipid mixture that created honeycombs resulted in a dramatic transformation of the assemblies, as evidenced by AFM micrographs (Fig. 4a-d): most of the molecules were no longer in honeycombs but aligned in strands of a few molecules in width. The assembly of the proteins is less compact and more dynamic, as shown in movie 1, making the individual pores of VDAC less visible by AFM, rendering most protein positions in the 2% Chol conditions appear loosely defined. To identify the position of the proteins in the membrane plane, we changed the parameters for protein identification and used three sets of conditions, which are either more or less stringent. This resulted in the image depicted in Fig. 4d, where a core of 35 proteins (blue) is robustly associated with all three collections. The two less stringent procedures yielded 63 (yellow) and 82 (magenta) proteins, respectively. The

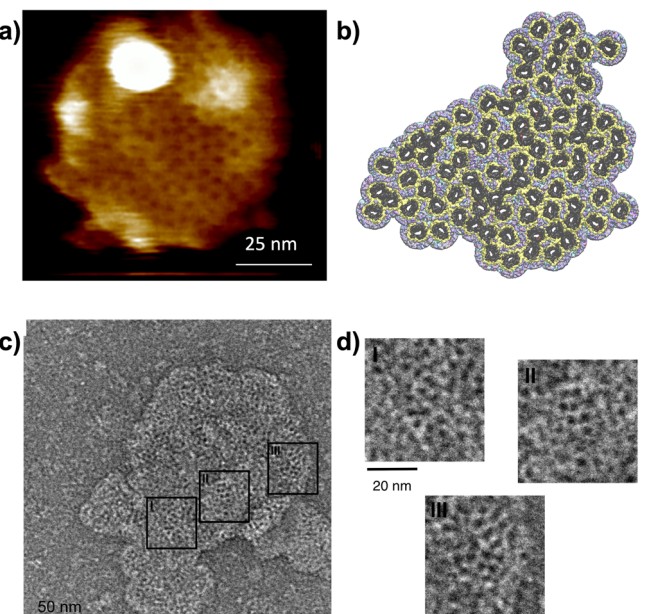

**Fig. 2 | VDAC1 forms honeycombs. a** AFM topography of VDAC1 in the POPC/POPE (65/35 w:w) membranes. All AFM measurements were performed at least in triplicate. **b** Computer model derived from AFM image in (**a**) (see Methods). VDAC1 is depicted as a dark solid surface, with the first lipid shell in yellow, and the other interstitial lipids in an atom color-coded representation. **c** Electron micrograph of outer membrane vesicles from *Neurospora crassa* mitochondria stained with uranyl formate at 68,000 × magnification, showing VDAC1 honeycomb-like assemblies. **d** The higher resolution of the three different areas from (**c**) shows that individual VDAC pores assemble as honeycombs.

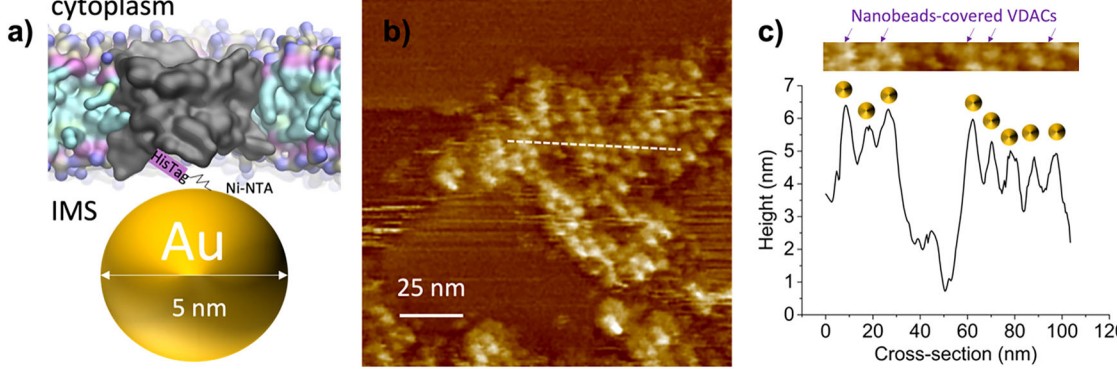

**Fig. 3 | VDAC1 forms parallel clusters. a** Schematic representation of a new method for orientation determination in AFM using gold nanoparticles coupled with Ni-NTA. The nanoparticles bind to the IMS-facing 6His-tag of VDAC1 and change the AFM profile. The distance between the bead and the His-tag is around 1.5 nm. **b** AFM imaging of honeycomb in the presence of gold nanoparticles reveals a parallel orientation of the molecules. **c** Height profile from (**b**) reveals peaks of the nanobead-covered VDAC, replacing the previous holes from the VDAC pores. Additional images are available in Fig. S4.

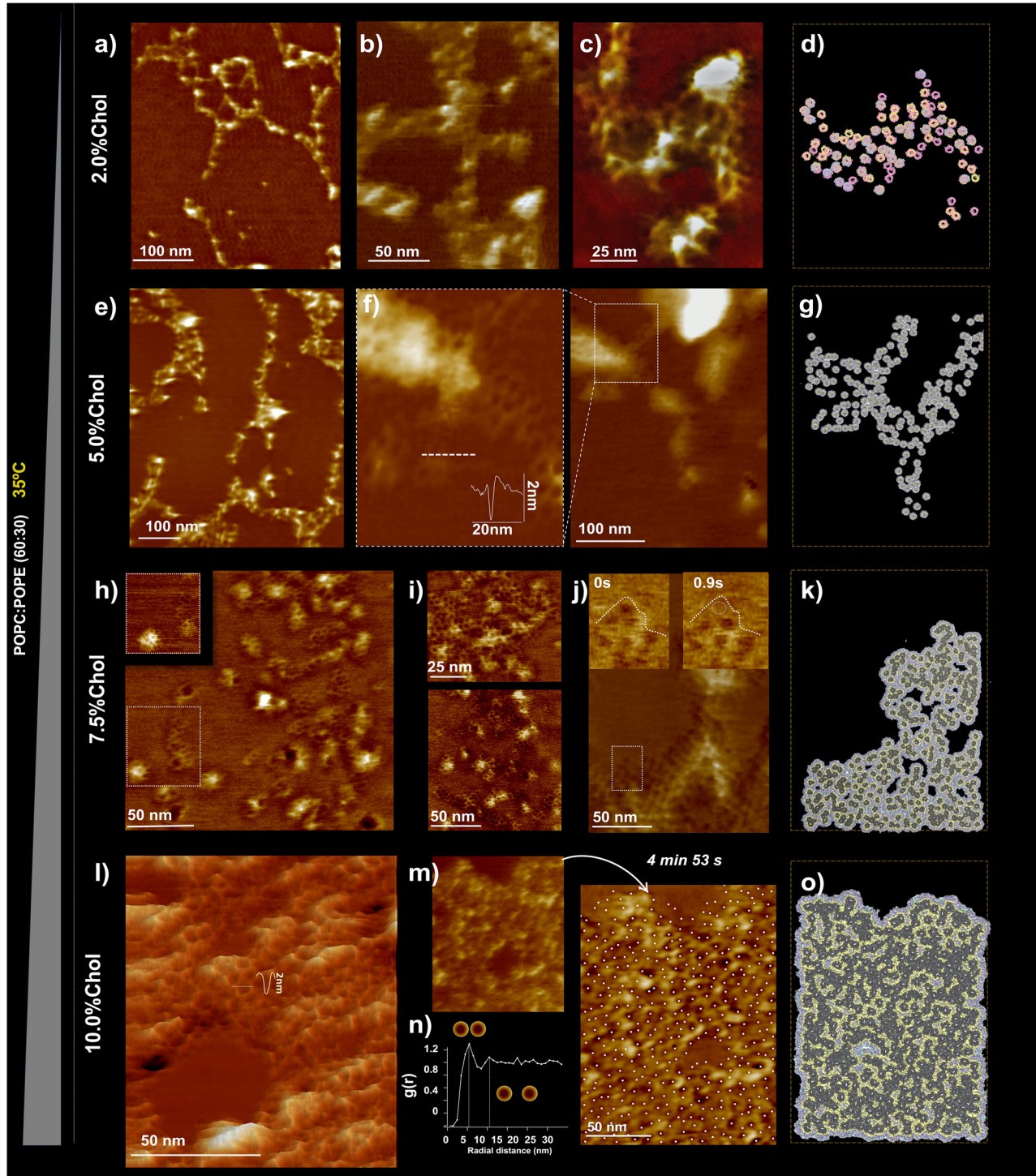

**Fig. 4 | Cholesterol controls the packing of VDAC1 in POPC/POPE bilayers (60:30).** AFM imaging at 33 °C (without gold nanobeads). All experiments were made in triplicate. The height of the false-color scale in all images was 11 nm. At 2.0% Chol (**a**, **b**, **c**), the assemblies are filament-like. **e** At 5.0% cholesterol, the assemblies are also filament-like, but significant honeycomb aggregations are observed. **f** Rows of VDAC are formed by one or a few pores in the transverse direction. **h**, **i** At 7.5% cholesterol, the honeycombs of VDAC are more prevalent. **j** Very rarely, an isolated pore moves at the border of the assembly, as shown in the inset. **l** At 10.0% cholesterol, the elongated shape of the VDAC arrangement is lost, and VDAC forms dense homogeneous assemblies with a rough topography. **m** The assemblies are stable as long as observed (several minutes) in all cholesterol conditions. **n** The interpore distance probability (radial distribution function) obtained from the positions

of the pores in a dense assembly localized by cross-correlation shows that the most probable distance between VDACs is 5.5 nm, and a second smaller peak is observed at 10.5 nm. *Right panel*: Models of supramolecular assemblies. The development of a coarse-grained molecular model from the various structures (**d**) from (**c**, **g**) from zoom in (**f**, **k**) from (**j**), and (**o**) from (**m**) allows us to accurately determine the context of protein–lipid interactions that support these distances. In (**d**), only VDAC molecules are represented, with blue corresponding to the positions with higher confidence, yellow and magenta, with two lower degrees of confidence, respectively. In (**g**, **k**, **o**), VDAC is depicted as a dark solid surface, with the first lipid shell in yellow and the other interstitial lipids in an atom color-coded representation. The higher resolution of these models is shown in Fig. S5.

superposition also illustrates the overall good alignment of the core proteins across the three distinct evaluations.

When the Chol content was increased to 5%, the observed VDAC1 structures progressively returned to the honeycomb organization, although with smaller and more linear honeycombs compared to those observed in the POPC/POPE membranes at 65/35 (Fig. 2a). Under these conditions, small protein patches of honeycomb structures resembling those imaged on native MOM[2] began to appear. When the Chol content increased to 7.5%, the elongation of VDAC assembly decreased, resembling native MOM conditions more closely. VDAC1 assemblies maintained a honeycomb structure, with more patches observed, similar to those seen at 5% cholesterol and in native MOM conditions. The assemblies became less elongated, and the compaction and extension increased. This trend is confirmed and more pronounced in PC/PE/10% Chol bilayers (Fig. 4l–o), where extended (up to several μm) and compact aggregates alternate with lipid areas. Within the honeycombs, small lipid patches were also observed: VDAC selected its ideal lipid–protein ratio, while leftover lipids accumulated in these patches. Overall, we conclude that although Chol is not essential for honeycomb formation, it significantly influences their assembly, particularly impacting their compaction and elongation.

### Physiological lipid ratios are essential for honeycomb formation

To further explore the impact of variable lipid ratios on VDAC1 organization, we tested different lipid compositions. In natural MOM, the lipid ratio PC/PE/Chol is ~60/30/10, and as already shown, this produces the honeycomb structure of VDAC observed on natural MOM (Table S1). Thus, the PC/PE/Chol 2% condition, forming small filaments of aligned VDACs, was beyond the physiological range, which is probably why it was not previously observed in the native MOM. Omitting Chol resulted in filaments and irregular lipid barriers of VDAC for a 50/50 mixture of POPC/ POPE (Fig. S6a) but allowed for honeycomb formation using close to a natural ratio of 65/35 (Fig. 2a). Omitting POPE, even at Chol concentrations up to 10%, failed to rescue honeycomb formation (Fig. S6b), indicating the necessity of POPE for such assemblies.

### Distances between the proteins illustrate the effect of cholesterol on compaction

Figure. 5 illustrates the VDAC–VDAC distances distribution within the structures under varying Chol (from 0 to 10%). For each cholesterol concentration, we categorized pairwise neighbor interactions into three types: direct protein–protein interactions (<53 Å), lipid-bridged interactions, and mixed lipid-protein-bridged interactions. As is clear in the AFM images, there are many fewer protein–protein contacts at 5% Chol (Fig. 4e–g) compared to the Chol-free sample, which corresponds to the marked decrease in the bar heights (Fig. 5) between the two concentrations, and a decrease in the compaction. Conversely, as Chol levels further increase, the process reverses and the assemblies become more compact (Fig. 4h–o), as evidenced by the overall rise in the bar heights in Fig. 5, especially in the direct protein–protein interactions (first two bars), reflecting enhanced compaction. As the compaction increases, the population of lipid-separated proteins is maintained or even reinforced (as shown in the next two bars). This suggests that multilayers of lipids persist between a significant fraction of the proteins, even in the dense regime with 10% Chol content. To further explore the heterogeneity of protein separation within the structures, we analyzed the distance distribution within smaller patches, specifically focusing on regions surrounding randomly selected proteins in the structures. The averages obtained from these ensembles (represented by the blue bars) are relatively close to those measured for the entire image. However, the standard deviation, along with the individual measurements (100, though 87 for the smallest structure), reveals considerable variation, indicating substantial heterogeneity within the structures.

Supplementary Fig. 5 presents the models at higher magnification to enhance the visualization of lipid roles. Discrete peaks observed in the Radial Distribution Functions (RDF) (Fig. S7) indicate peaks separated by 6 Å,

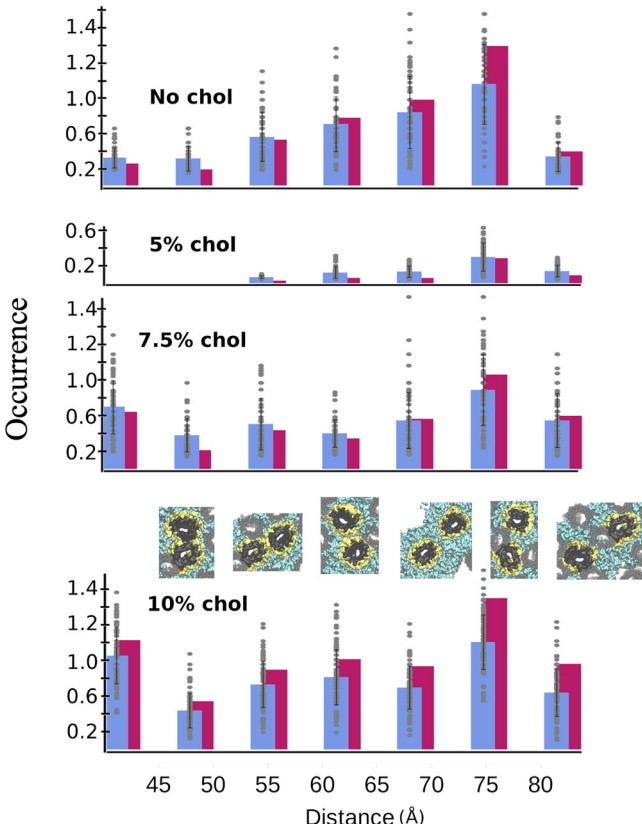

**Fig. 5 | Reversible cholesterol-driven compaction of VDAC1 assemblies.** The distance between the VDAC particles was obtained from the AFM micrographs. A context was assigned based on the computed assembly models, including strong (clashing) [<50 Å] and soft direct contact [50–56 Å], and successive contexts of lipid-separated proteins from [56–62 Å], to [62–68 Å]. More distant relationships between proteins include lipid or protein bridged distances [68–80 Å] and row [80–86 Å]. At each distance, protein separation may result from a mixture of lipid- and protein-mediated interactions following local protein density. The magenta bars represent the distributions observed across entire structures, while the blue bars show results from a similar analysis performed on smaller patches, each centered on an individual VDAC protein (see Methods and ref. 63). Individual patch results are shown as gray dots (with *n*, the number of observation within each structure, see Supplementary Data 2 for "*n*" value at each condition). The error bar is the standard deviation calculated over the grey dots for each context and condition, which quantifies the heterogeneity within each structure. Factors such as the overall assembly size and consequently the proportion of proteins located at the border contribute to this observed heterogeneity. Model vignettes were extracted from the molecular models corresponding to the given distance measurements: VDAC molecules considered for the distance measurement are in dark grey, the first lipid shell in yellow and additional lipids in cyan. The occurrences represent the average number of proteins present at a specified distance.

which represent distinct lipid arrangements in lipid-bridged protein separations, as illustrated in Supplementary Fig. 3.

To further analyze protein arrangement within the various structures, we examined their internal configuration in terms of clusters. These clusters are defined as groups of proteins connected by protein–protein interactions within a distance of 5.6 nm. Our cluster analysis (see Methods and Table 1) indicates that the fraction of monomers (fmono) is the most variable feature. This fraction ranges from predominantly monomeric in the 5% Chol condition, where almost all VDAC molecules are completely surrounded by lipid, to only 0.05 in the most aggregated 10% Chol condition. Notably, even under this densely packed condition, a significant portion of the molecules remains separate, including some located in the core of the assembly (see

**Table 1 | Distribution of VDAC within the structures, as a free monomer or among clusters**

|  | no Chol | 2% | 5% | 7.5% | 10% |
|---|---|---|---|---|---|
| # VDAC in structures | 87 | 63 | 252 | 330 | 399 |
| (fmono) | 0.45 | 0.56 | 0.98 | 0.31 | 0.05 |
| # Clusters | 16 | 10 | 2 | 65 | 14 |
| (fclust) | 0.55 | 0.44 | 0.02 | 0.69 | 0.95 |
| <Clust size> | 3.08 ± 0.17 | 2.7 ± 0.09 | 2.0 ± 0 | 3.54 ± 0.16 | 38.38 ± 27.59 |

# VDAC represents the number of proteins identified in our reference AFM images for the various structures depicted in Fig. 2a and 4c, f, j, m, which are separated into monomers and clusters of different sizes. (fmono) denote the fraction of VDAC molecules categorized as isolated particles using our DBSCAN algorithm (see Methods and ref. 30 for the program). # Clusters indicates the total number of clusters identified. (fclust)denotes the fraction of VDAC molecules within those clusters. Finally, <Cluster Size> refers to the average number of VDAC calculated across all clusters (excluding isolated monomers).

**Table 2 | Lipid composition of the MD simulations of Simple Membrane 1 (SM1) and Simple Membrane 2 (MS2)**

| Lip | #lip tot | Per leaflet | Mass (Da) | %W | %mol |
|---|---|---|---|---|---|
| **SM1** | | | | | |
| POPC | 716 | 358 | 536,747 | 66.4% | 65.1% |
| POPE | 384 | 192 | 272,111 | 33.6% | 34.9% |
| **SM2** | | | | | |
| POPC | 642 | 321 | 481,622 | 62.1% | 58.4% |
| POPE | 345 | 172 | 244,164 | 31.5% | 31.4% |
| CDL2 | 6 | 3 | 8532 | 1.1% | 0.5% |
| CHOL | 107 | 54 | 41,205 | 5.3% | 9.7% |
| Tot | 1100 | | 775,524 | 100% | 100% |

supplementary Fig 8). The protein organizations are consistently a mixture of multimeric and monomeric proteins (Fig. S8).

Additionally, the overall compactness of the structures can be assessed through the fraction of proteins in oligomers. A value of 0.95 further demonstrates the highly aggregated nature of the 10% Chol condition. Furthermore, it also reveals that the 7.5% Chol condition is more compact than the conditions with 2% or no cholesterol because it demonstrates fewer monomers and nearly twice as many oligomers.

To further emphasize the lack of compactness observed in the various structures, we calculated the average number of VDAC proteins within oligomeric clusters, which typically ranges from two to three. This highlights the mosaicism of these structures, with many small oligomers, until the Chol content reaches 10%, at which point the average increases to 27.

In fact, in the various structures, only about 10% of the proteins contribute to clusters larger than 10 molecules. The intermixing of small assemblies with a relatively high content of free monomers, combined with the presence of numerous lipids, often leads to the formation of stretched rather than compact clusters, resulting in a porous or spongy assembly (Fig. S8).

## VDAC1 has different affinities for MOM lipids

To understand VDAC1–lipid interactions, we conducted MD simulations of monomeric VDAC1 in membranes approximating the composition of the MOM using only two (SM1) or three (SM2) lipids: POPC, POPE, and Chol. We called them simple membrane 1 (SM1, POPC/POPE at 65/35 (w:w)) and simple membrane 2 (SM2, POPC/POPE/Chol/CL at 62.5/31.5/5.3/1.1 (w:w)), which precise compositions described in Table 2 in the method section. Despite initial random lipid distributions, convergence occurred across three independent MD simulations for each system on the microsecond timescale. Analysis of lipid positions over the final 10 μs revealed consistent distributions across replicates (Fig. S9). The probabilities of the presence of each lipid from the simulations are shown in Fig. 6.

The MD simulation reveals that VDAC influences the local lipid distribution within the membrane, creating distinct lipid layers. Chol and PE are preferentially located in the first lipid layer in direct contact with VDAC, while PC is more prevalent in a second, adjacent layer. This creates an uneven (anisotropic) distribution of lipids around VDAC1, despite the relatively small structural differences between the head groups of PC and PE. The coarse-grained model emphasizes how these subtle head group differences influence lipid–protein interactions and result in this distinct spatial arrangement. This anisotropic distribution of lipids extended radially to the bulk, demonstrating long-range effects (at least 5 nm from the protein surface), as observed for other proteins[30]. Adding Chol and trace amounts of CLs to SM1, we created SM2. On the timescale of our study (20 μs), the distribution of CLs was meaningless and suffered from poor convergence owing to its low concentration. In contrast, the behavior of Chol showed

strong convergence (Fig. 5d, e), challenging the role of POPE in the proximal layer, with less impact on PC densities.

In both SM1 and SM2, the upper leaflet (cytosolic side) predominantly accommodated sites for interactions between PE and Chol, while the IMS leaflet exhibited a preference for PC lipid concentration (Fig. 5c, f; the side view of lipid distribution is asymmetric between membrane leaflets). Notably, both leaflets contained equal numbers of lipids. We conclude that VDAC1 preferentially interacts with Chol and POPE, positioning POPC in a secondary layer. This arrangement results in a distinct lipid organization around VDAC1 within the membrane.

## Discussion

VDAC organization is critical for various aspects of cellular physiology, however, little is known about what controls the interactions that drive VDAC organization in the membrane and how these interactions depend on the lipid environment. We have examined how the organization and dynamics of VDAC1 within a lipid bilayer are modulated by PE and Chol, offering insight into the rules underlying membrane protein interactions and assembly into large-scale structures.

Our results on VDAC assembly in POPC/POPE membranes in the absence of Chol reveal the formation of disordered, glassy structures enriched with lipids[28], which we refer to as "honeycombs." The images we obtained are strongly reminiscent of those observed in isolated MOM[1,2]. Even in the absence of Chol, VDAC forms protein–lipid honeycomb assemblies, although these are fewer in number and smaller in size compared to those observed with 5% or more cholesterol. All honeycomb assemblies exhibit high stability, with lipids acting as molecular "mortar" to stabilize VDAC-lipid assemblies, similar to lipopolysaccharide-mediated interactions in E. coli[31]. Electrostatic or other short-range intermolecular interactions alone cannot fully account for this stability, as these are typically confined to the immediate vicinity of membrane proteins. Notably, we observed Chol-induced changes capable of producing long-range effects, including phase transitions. Such loosely ordered, noncrystalline organization is indicative of long-range (5–10 nm) effective interaction potential between proteins of intermediate strength (interactions in the range of –0.5 to –5 kJ/mol)[32]. We have previously examined in MD simulations how such long-range interactions can be lipid mediated[30]. In the current study, MD simulations revealed that under these conditions, VDAC forms a surprisingly isotropic lipid annulus. This annulus exhibits a clear preference for PE lipids adjacent to the protein, while PC occupies the second solvation shell. This preference may stem from the smaller headgroup of PE compared to PC, which could facilitate tighter packing near the protein surface.

The addition of a small percentage (2%) of Chol dramatically alters the observed structures. The honeycomb-like assemblies are replaced by more filamentous assemblies (Fig. 4a) that exhibit considerable dynamics. The formation of these chain-like structures suggests that the underlying protein–protein interactions become anisotropic[33], with specific preferences for particular surfaces of VDAC to interact. MD simulations in the presence

**Fig. 6 | VDAC influences the local lipid distribution within the membrane.** Volumetric maps representing the individual lipid occupancies computed from molecular dynamics simulations of VDAC1 averaged over $3 \times 1000$ frames (1000 per replica, see Fig. S9 for the individual maps). Volumetric maps are not direct representations of lipid densities but rather reflect the probability of the presence of each lipid relative to its number. The densities are shown at a cut-off level corresponding to either 1.2 or 1.8 fold (enrichment) of the average density of each individual lipid. The enrichment of each lipid around VDAC1 is calculated relative to the average density of each individual lipid. The upper row (**a**, **b**, **c**) shows the data for the simple membrane 1 (POPC/POPE 65/35) and the lower row (**d**, **e**, **f**) is for the simple membrane 2 (POPC/POPE/Chol/CL 62.5/31.5/5.3/1.1). The mean occupancies of POPC (blue), POPE (magenta) lipids, and cholesterol (gold) were calculated over the last 10 µs of three 20 µs coarse-grained molecular dynamics simulations, the upper rows show the averaged map corresponding to the three distinct simulations of SM1. The lower row represents a similar extraction for SM2. The top view is from the cytosol. Black spheres indicate the position of E73, a charged amino acid, facing the membrane. The cytosolic side accommodates sites for interactions between PE and cholesterol, while the intermembrane space leaflet exhibits a preference for PC lipids.

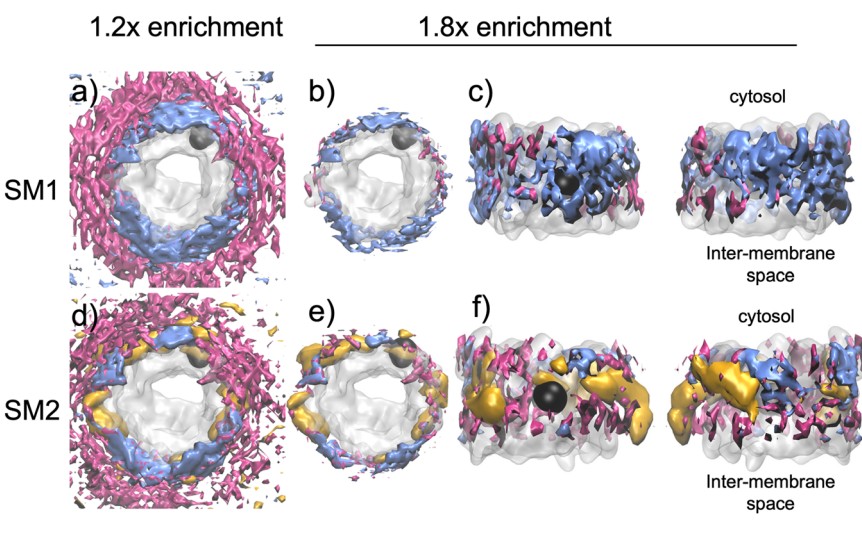

of Chol (Fig. 6), combined with previous experimental analyses[26,34], reveal the existence of multiple Chol binding sites on the VDAC surface. We hypothesize that at low cholesterol concentrations, only one or two of these binding sites are occupied, leading to the formation of an anisotropic protein-lipid complex. Under these conditions, most of the VDAC surface shows a preference for PE lipids, while cholesterol selectively occupies a few specific sites, driving the assembly of dynamic and filamentous assemblies.

The addition of more Chol (5–10%) lead to the gradual reappearance of the honeycomb-like structures (Fig. 4 and Fig. S5). The resulting assemblies grew progressively larger, more compact, and exhibited reduced dynamics as the Chol content in the membrane increased. MD simulations suggest that, under these conditions, an increasing amount of cholesterol associates with the VDAC surface (Fig. 5), promoting the formation of a more isotropic protein-lipid complex that favors the reassembly of the honeycomb assembly. We suspect that a part of this preference of Chol for the protein surface, beyond the presence of cholesterol binding sites, is due to the hydrophobic mismatch between VDAC and the lipid membrane. Thus, the small hydrophobic width of VDAC compared to the hydrophobic width of the membrane (Fig. S10) results in the selection of Chol, which is significantly smaller than POPE or POPC. Of course, cholesterol also has a very small headgroup, so this probably also participates in the selection of cholesterol over the other lipids in the membrane.

MD simulations reveal VDAC1's strong affinity for POPE, with an identified binding site[16]. Affinity depends on local lipid concentration; hence, lipid ratios are important and can explain the presence of honeycombs at PC:PE (65:35) and not (50:50). As we did not observe any glass-like structure[28] of stable VDAC1-lipid organization using only POPC and Chol, we conclude that in our simplified three-lipid system POPE is necessary for honeycomb formation and cannot be replaced by cholesterol, even at higher concentrations.

An interesting observation is the variation in compaction of assemblies and their internal structure with Chol concentrations (Fig. S5, 8 and Table 1). Analysis of the internal organization of the aggregates showed that there are isolated "monomeric" VDAC molecules that do not associate with

their nearby neighbors, and there are multiple indirect protein–protein contacts with intercalated lipids. This implies that there can be a significant energy barrier to excluding lipids from an interface in agreement with previous observations.

We propose that, at a molecular level, this effect of Chol on compaction is a result of the small size and reduced dynamics of Chol. Unlike diacylglycerol-based lipids, Chol's rigid, polycyclic structure limits its flexibility, thereby decreasing the dispersive pressure it exerts on neighboring proteins. In contrast, acyl chain-containing lipids, when adapting to hydrophobic mismatch, may enhance dispersive forces between proteins, further emphasizing the distinct role of cholesterol in promoting compaction.

Thus, at the molecular level, our results can be interpreted in the context of lipid selection by proteins and lipid-mediated long-range interactions in driving the assembly of distinct structural types. Proteins, such as VDAC, selectively recruit lipids to form a lipid annulus that can be more or less isotropic, with the specific lipid composition modulating both the strength and the spatial extent of protein–protein interactions. We observed that PE and Chol are key regulators of VDAC clustering. In our experiments, the emergence of different structural assemblies occurred within physiological Chol concentration ranges. This suggests that in a cellular context, different regions of the MOM or distinct cellular states may dynamically influence VDAC organization, enabling the formation of varied assemblies over time.

We identified diverse organizations of VDAC1, primarily characterized by a honeycomb pattern, consistent with previous findings on native membranes[1,2] and mitochondrial vesicles from *Neurospora crassa* (Fig. 2c, d). The heterogeneity and lipid-rich nature of these honeycombs likely contribute to the multitude of oligomeric forms identified after detergent solubilization from native sources[35]. The diverse inter-protomer interaction areas potentially result in the dissociation of honeycomb assemblies and the preservation of only direct protein–protein interactions.

VDAC1 has been observed to crystallize as an antiparallel dimer in DMPC lipids[36,37]. In contrast, we observed parallel assemblies of VDAC1

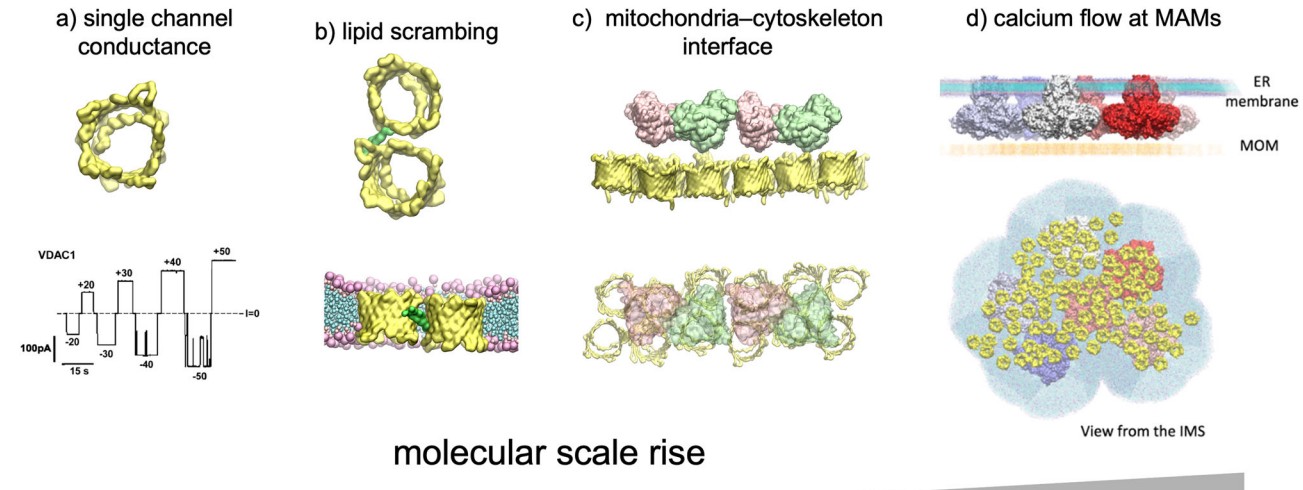

**Fig. 7 | Lipid-induced oligomerization of VDAC can be associated with different biological events. a** Single channel can conduct ions and metabolites. **b** Jahn et al.[9] identified a specific VDAC dimer interface involving strands β1,2 and 18–19 on the second protomer involved in lipid scrambling. In the honeycombs, we identified many protein–protein contacts that could serve to scramble lipids. **c** Arrays of VDAC serve as docking platform for glycerol kinase in sperm mitochondria (PDB ID: 7NIE)[46]. **d** Model of the IP3R-VDAC complex implicated in calcium flow. Bartok et al.[45] demonstrated the presence of clusters of IP3R at ER-mitochondria contact sites, forming direct contacts with VDAC. As VDAC is thinner than the membrane, only proteins are depicted in the MOM.

within the honeycomb assembly (Fig. 3 and Fig. S 4). We propose that the lipid asymmetry observed between the membrane leaflets in our MD simulations (Fig. 6) compensates for protein asymmetry, facilitating parallel packing.

The sensitivity of VDAC1 oligomeric organization to lipid composition has significant physiological implications, as MOM lipid composition can vary substantially owing to interactions with neighboring membranes, such as mitochondrial inner membrane/MOM or endoplasmic reticulum (ER)/MOM contact sites, facilitating lipid transfer. Additionally, changes in MOM lipid composition occur under various pathological conditions; for instance, cancer cells exhibit altered lipid metabolism and membrane composition[18]. The dysregulated accumulation of Chol in mitochondrial membranes influences mitochondrial permeability during cell death induction by Bax and death ligands[25,38]. In neurodegenerative conditions such as Alzheimer's disease, a change in mitochondrial lipid profiles is also observed[19], including increased mitochondrial Chol levels[20–22], which is particularly relevant given VDAC1's involvement in Alzheimer's pathology[39]. Therefore, our findings suggest that lipid variation could serve as an effective mechanism for regulating VDAC organization and function.

While VDAC is primarily known as an ion channel, its oligomerization's effect on gating remains debated. We found that 10% Chol triggers very dense VDAC organization, but even at 25% cholesterol, VDAC1's conductance and gating remain unchanged[15]. This suggests that honeycomb formation enhances ion and metabolite flow by increasing VDAC density without altering single-channel properties.

VDAC oligomerization is implicated in crucial physiological processes, notably in calcium, metabolites and lipids homeostasis and apoptosis[13], but also in pathological conditions such as diseases like Alzheimer's or lupus[8]. During apoptosis, the increased VDAC oligomerization identified by in-cell cross-linking[13] is compatible with the honeycomb assembly characterized in our study, particularly favored by increased Chol. VDAC recruits pore-forming proteins Bax and Bak, which are responsible for MOM permeability[40,41]. Therefore, such oligomerization could serve as a platform for the recruitment of these partners and subsequent pore formation in the MOM. Supporting this hypothesis, some models suggest that Bax and Bak bind to multimeric VDAC complexes[42].

Mitochondrion-associated membranes (MAMs), where the ER and mitochondria interact, serve as key hubs for lipid and calcium transport.

Chol-rich MAMs[24] likely favor large VDAC arrangements, which can enhance their efficiency as a lipid scramblase[9]. VDAC oligomers also facilitate calcium flux from the ER to mitochondria, with the inositol 1,4,5-triphosphate receptor (IP3R) in the ER membrane releasing calcium that passes through VDAC in the MOM[43,44]. IP3R forms large mega-Dalton tetrameric complexes, in direct contact with VDAC[45], releasing large amounts of calcium into the IMS, which is channeled through VDAC into the MOM via passive diffusion. To provide efficient calcium transfer, a large number of VDAC molecules are necessary. Fig. 7d illustrates the stark size difference between VDAC (~30 kDa) and the much larger IP3R complexes (megadalton), highlighting the need for densely packed VDAC assemblies at MAMs (mitochondria-associated membranes) to ensure efficient calcium uptake by mitochondria. VDAC oligomers play a role in sperm mitochondria, as shown by Leung et al. using cryo-electron tomography, where VDAC is used as a platform to recruit glycerol kinase, serving to tether neighboring mitochondria together[46].

VDAC oligomerization and its intricate interaction networks are critical for various physiological processes; however, the forces driving VDAC oligomerization remain poorly understood. Building on prior studies[1,2,23], our findings demonstrate that VDAC1 assembly is strongly modulated by lipid content, providing a mechanism for its dynamic reorganization within the MOM under changing conditions. VDAC can adopt diverse oligomeric states in the MOM[1,2], fulfilling distinct physiological roles (Fig. 7): from single-channels that conduct ions and metabolites and recruit partner proteins (e.g., tubulin or α-synuclein[12,47]), to small oligomers like dimers that mediate lipid scrambling, and larger assemblies that serve as platforms for protein tethering or efficient calcium flux. Regulation by local membrane lipid composition enables the spatial formation of distinct VDAC assemblies within specific MOM regions. Future studies involving different lipids and different VDAC isoforms, as well as at mitochondrial and cellular scales, will be crucial to further elucidate this dynamic process.

## Methods
### VDAC1 production and purification
A pQE9 plasmid with the mouse VDAC1 gene bearing a 6-his tag in N-terminus was produced in inclusion bodies in *Escherichia coli: M15 (pREP4)* (competent cells Creative Biolabs) cells[48,49]. Cells were grown at 37 °C under agitation until OD of 0.7, and the protein production was induced for 4 h with 1 mM IPTG. Cells were harvested by centrifugation

and resuspended in buffer (50 mM Tris pH8, 2 mM EDTA, 20% sucrose), and lysed with lysozyme and sonication. The VDAC-containing inclusion bodies were isolated by a 15 min centrifugation at $12,000 \times g$. The inclusion bodies pellet was washed with 20 mM Tris pH8, 300 mM NaCl, 2 mM CaCl$_2$ and solubilized in 20 mM Tris pH 8, 300 mM, 8 M urea for 1 h. Insoluble material was removed by centrifugation at $25,000\,g$ for 1 h. mVDAC1 contained in the supernatant was purified by Ni-NTA metal affinity column, and eluted with 20 mM Tris pH8, 300 mM NaCl, 8 M urea, 150 mM imidazole. The pure mVDAC1 was refolded by rapid dilution in buffer 20 mM Tris pH8, 150 mM NaCl, 10 mM DTT, 1.5% LDAO. The refolded protein was ultracentrifuged at $355,000\,g$ for 45 min to remove aggregates, concentrated, and injected on a Superdex 200 increase 10/300 GL column (Fig. S11). The monomeric VDAC peak was collected and concentrated to 2 mg/ml. Stocks were flash-frozen in liquid nitrogen and stored in 20% glycerol at $-80\,°C$. Samples were ultracentrifuged after thawing before reconstitution into liposomes to remove any possible aggregates.

## Reconstitution into liposomes

POPE, POPC, and Chol lipids were purchased from Avanti polar lipids in chloroform, mixed at the desired ratios and dried under nitrogen gas. The resulting lipid film was resuspended by vortexing in 150 mM KCl, 20 mM Tris pH 8 to a final concentration of 12.5 mg/ml of lipids. The suspension was heated at 37 °C for 10 min then sonicated for 1 min or until formation of small unilamellar vesicles (100 nm), as measured by dynamic light scattering (DLS). Detergent β-OG was added to a 0.65 detergent/lipid ratio (w:w). Proteins were added to the destabilized liposomes to a 3:1 lipid:V-DAC ratio (w:w). Detergent removal was achieved by addition of Bio-beads at a Bio-bead/detergent ratio of 30 (w:w) overnight. The liposomes were analyzed by DLS after reconstitution.

## AFM and HS-AFM data acquisition

To form a homogeneous protein containing supported lipid bilayer (SLB), a mixture of 15 μl of the proteoliposome suspension (1 mg/ml) and 85 μl of "Incubation buffer" (10 mM HEPES pH 7.4, 100 mM NaCl, 15 mM CaCl$_2$) was incubated 1 h at room temperature on a freshly cleaved mica under a humid hood. The sample was rinsed 10 times in ultrapure water (MilliQ) and imaged by AFM/HS-AFM in "Imaging buffer" (10 mM HEPES pH 7.4, 100 mM NaCl, 5 mM CaCl$_2$) between 33–35 °C to check the homogeneity of the formed membrane. All conditions have been performed in triplicates.

Conventional AFM imaging was performed in contact mode with a Nanoscope IIIe Multimode AFM (Bruker, Santa Barbara, CA, USA) using OTR4 probes of $k = 0.1$ N/m. The scan rate, gains, and deflection setpoint were adjusted during acquisition to optimize the quality of acquired images.

HS-AFM movies were acquired in amplitude modulation mode optimized high–resolution imaging parameters with a modified HS-AFM (SS-NEX, RIBM, Tsukuba, Japan)[50]. The apparatus is equipped with our custom-made digital high-speed lock-in amplifier coded on a reconfigurable FPGA using LabView (National Instruments, Austin, USA). Short cantilevers (length~7 μm) designed for HS-AFM, presenting an electron beam deposition tip, were used (USC-F1.2-k0.15 Nanoworld, Switzerland). They are characterized by a nominal spring constant $k = 0.06$–$0.3$ N/m, a resonance frequency in liquid of fres = 500 kHz and a quality factor Qc~2 (in liquid). HS-AFM sensitivity to probe the deflection was 0.1 V/nm. The imaging amplitude setpoint was set to ~90% of the free amplitude (~1 nm). All experiments were performed between 33 and 35 °C, by introducing the HS-AFM microscope inside a temperature-controlled box.

## Image analysis

Acquired images were plane-fit and flattened using the AFM processing and analysis software provided by the instrument manufacturer. HS-AFM image treatment was limited to the correction of XY drift and a first-order X-line fit. Image analysis was performed using the general distribution of ImageJ[51] and WSxM[52].

To study the particle distribution, the computation of the RDF was implemented. This measure is obtained by computing the particle density in a ring of radius $r$ and thickness $dr$ for each particle and then averaging. The RDF is calculated based on the geometric centers of the VDAC circles identified in the AFM images, with pixels corresponding to the same VDAC excluded from the analysis.

### Radial distribution. Mathematical definition:

$N$ is defined as the number of holes selected in a patch and $P_i$ as the position of the hole $i$. $S_{patch}$ is the concave or convex surface including all the positions dilated with a given radius. The density of points in the entire patch surface is $\rho = \frac{N}{S_{patch}}$. The RDF of a reference particle is the density of particles at a given distance from it. In order to calculate it, it is usually necessary to count the number $N_i(r, dr)$ of particles that exist within a distance $r - \frac{dr}{2}$ and $r + \frac{dr}{2}$ from the reference particle. The RDF $g(r)$ corresponds to this value, $N_i(r, dr)$ normalized by the global density multiplied by the ring surface ($S_{ring}(r, dr) = 2\pi r dr$). Thus $g(r) = \frac{1}{N}\sum_{i=1}^{N} \frac{N_i(r,dr)}{\rho S_{ring}}$. However, this approach requires discarding particles at a distance $r$ from the boundaries, which in our case affects the statistics due to the small number of points in a patch. To solve this predicament, instead of normalizing by the full ring, we normalize by the intersection of the ring and the patch surface (Fig. S12).

$$g(r) = \frac{1}{N}\sum_{i=1}^{N} \frac{N_i(r, dr)}{\rho S_{ring} \bigcap S_{patch}}$$

The Code to calculate the RDF is accessible here: https://github.com/CodeCanSolveAll/Particle_Analysis

## Coarse-grain setup and molecular dynamics simulations

The coarse-grain structure of hVDAC1 was modeled based on that of mouse VDAC1 (at higher resolution than hVDAC1, PDB ID: 3EMN)[37], which differs by only four conservative mutations over 254 residues. First, the homology modeling was set up using MODELLER[53]. Then the protein was embedded in membranes using the CG membrane builder[54] tool from CHARM-GUI[55] using the MARTINI 2.2 force field[56] with elastic networks[57] using an elastic force constant of 500 kJ/mol/nm² between all backbone particles falling below 0.9 nm distance. This setup allowed for the production of stable VDAC conformations throughout the 20 μs simulation, as evidenced by the RMSD time series shown in Fig. S13.

Two distinct membrane compositions (Table 2) were build based on the main lipids of the MOM membrane[27] (Table S1). The first one, Simple Membrane 1 (SM1), is made only of POPC and POPE, and the second one, Simple Membrane 2 (SM2), contains the same two principal lipids plus 10% molar Chol and a trace amount of CL. The final simulation boxes comprise 1100 lipids randomly distributed within the membrane plane. The periodic rectangular boxes also contain 13,000 CG water particles and 100 mM NaCl. For replication purposes, three distinct systems corresponding to alternative random lipid distributions were built for the two membrane systems. The various systems were heated (to 303.15 K), equilibrated, and run over 20 μs using the default protocol and parameters provided by CHARMM-GUI. Briefly, the simulation protocol consisted of sequential minimization, equilibration, and production phases. The system was first minimized using two steps: (1) a soft-core minimization (5000 steps) followed by (2) a steepest descent minimization (5000 steps). Both minimization steps used a Verlet cutoff scheme with reaction-field electrostatics ($\varepsilon_r = 15$) and cutoffs for both Coulomb and van der Waals interactions set to 1.1 nm. The equilibration phase comprised five sequential steps with progressively decreasing harmonic position restraints on both the protein backbone and lipid head groups: initial equilibration with strong position restraints (1000 kJ·mol⁻¹·nm⁻² for proteins and 200 kJ·mol⁻¹·nm⁻² for lipids), followed by sequential reductions to 500, 250, 100, and 50 kJ·mol⁻¹·nm⁻² for the protein and 100, 50, and 20 kJ·mol⁻¹·nm⁻² for lipids.

Throughout the equilibration, the timestep was gradually increased from 2 to 20 fs. Temperature coupling was maintained at 303.15 K using the

v-rescale thermostat ($\tau = 1.0$ ps) for protein, membrane, and solute groups. Semi-isotropic pressure coupling was implemented using the Berendsen barostat ($\tau = 5.0$ ps, compressibility = $3 \times 10^{-4}$ bar$^{-1}$) at 1.0 bar.

The production phase was conducted using a 20 fs timestep for 20 µs (1 billion steps). The Parrinello-Rahman barostat replaced the Berendsen barostat ($\tau = 12.0$ ps) while maintaining semi-isotropic pressure coupling. System coordinates, velocities, and forces were saved every 200,000 steps, with compressed trajectory output every 50,000 steps. The temperature and pressure coupling parameters remained consistent with the equilibration phase.

The simulations were run on the supercomputer facilities provided by the GENCI at the CINES—Montpellier, France—or at the TGCC (CEA)—Saclay, France using GROMACS v2018[58].

## Volumetric map analysis
The 20 µs simulation trajectories were aligned to a reference configuration, which shared the same reference frame as the original all-atom initial structure. The Volmap tool from VMD was used to generate 3D histograms of particle counts across different time intervals of the simulation replicas. The averaged particle count (occupancy) was directly binned on a 2 Å resolution grid, without applying any additional Gaussian averaging. While parallel evolution of lipid densities was observed across replicas starting from 5 µs, our analysis focuses on the final 10 µs of each simulation to ensure convergence between the replicas.

## Building of model assemblies
The coordinates extracted from the AFM micrographs were used to position proteins and lipids extracted from MD configurations of the monomeric systems. Distinct configurations were randomly chosen from the last 10 µs of the various MD simulations. The proteins were extracted with a cylindrical extrusion of lipids with a radius of 5 nm about the protein center. Since the experimental orientation of the individual proteins is unknown, we randomly rotated the proteins and their associated lipids around the channel axis, which is perpendicular to the membrane plane. An optimization procedure was developed to select orientations that minimize clashes between proteins. In this Python program, overlapping lipids from the added configurations were sequentially removed from the model. The resulting stereochemical models were used to evaluate lipid–protein and inter-protein contacts in order to gain insights into the assemblies at the molecular level[59].

## Protein distance distribution
The number of proteins at various proximities was calculated using the integral of the $g(r)$ function to determine the average number of proteins within different concentric shells surrounding a reference protein. The results were organized based on the different classes illustrated in Fig. 5 by summing the corresponding shells. To assess the heterogeneity of the various aggregates, the $g(r)$ calculation was repeated on smaller patches within the overall aggregates. These patches were defined by selecting proteins within 15 nm of a randomly chosen protein. The $g(r)$ integral was computed from 0 to 10 nm for all selected proteins. For the condition with 5% Chol, which is known as less densely packed proteins, this range was extended to 50 nm to encompass the broader area represented in the wide field of the image. This procedure was repeated 100 times for all conditions, except for the "no Chol" condition, which included only 87 proteins.

## Cluster analysis
An analysis of particle aggregation was performed using the datasets obtained from the AFM particle coordinates. Clustering was executed with the Density-Based Spatial Clustering of Applications with Noise (DBSCAN) algorithm[60], implemented in Python using the "scikit-learn" library. The DBSCAN algorithm was applied to a distance matrix generated by calculating the pairwise distances between all particles.

The algorithm was configured with a radius (epsilon) of 5.3 nm, which enabled the extraction of clusters that could be visually validated against the previously described coarse-grain model of assemblies. Additionally, a minimum sample size of 2 was set, indicating that at least two particles (a dimer) are required to form a cluster.

## Screening of outer membrane vesicles by negative staining microscopy
Outer membrane vesicles (OMVs) isolated from *Neurospora crassa* mitochondria were kindly provided by Prof. Stephan Nussberger (see ref. 61). Frozen OMVs were thawed and applied onto copper grids (400 mesh) backed with a continuous carbon layer, and subsequently stained with uranyl formate. Data was collected using a Tecnai Spirit Biotwin electron microscope at 120 kV (FEI), fitted with a CCD camera (Gatan). Micrographs were acquired with 2 s exposure at 49,000 ×, 68,000 ×, and 76,000 × nominal magnification.

## Statistics and Reproducibility
All the experiments, of AFM or of molecular dynamic simulations, are replicates of 3 independent experiments.

## Reporting summary
Further information on research design is available in the Nature Portfolio Reporting Summary linked to this article.

## Data availability
The data that support this study are available in Supplementary Data 1.

## Code availability
Input files and data for model building and RDF calculations can be accessed at https://github.com/CodeCanSolveAll/Particle_Analysis. Programs for analyzing the images can be found at https://github.com/deejy/ComBiol2025[59,62,63].

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

## Acknowledgments
We thank Prof. Werner Kühlbrandt and Prof. Stephan Nussberger for sharing the negative stain images. This work was supported by the Aix-Marseille Université grant AMX-21-PEP-022 (L.B.), PhD fellowships from Aix-Marseille Université (E.L. and V.R.), and French grants ANR-21-CE42-0031 and ANR-19-CE11-0024 (I.C.). This work was performed using HPC resources from GENCI–IDRIS (Grant 2020-A0080707044 and 2021-A0110707044).

## Author contributions
Conception: L.B., J.P.D., and I.C. Sample preparation: L.B., V.R.; AFM imaging: E.L., N.B.; Computational and molecular modeling: J.P.D.; Data analysis: J.P.D., E.L., A.O., J.S., I.C., N.B., L.B.; Negative stain: P.O., L.B., and J.P.D. wrote the manuscript. All the authors reviewed it.

## Competing interests
The authors declare no competing interests.
