## [Transparent Peer Review file · Communications Biology]

Membrane Lipid composition modulates the organization of VDAC1, a mitochondrial gatekeeper

Corresponding Author: Dr Lucie Bergdoll

This manuscript has been previously submitted at another journal. This document only contains information relating to versions considered at Communications Biology.

Version 0:

Reviewer comments:

Reviewer #1

(Remarks to the Author)

In the manuscript "Lipids modulate the organization of VDAC1, the gatekeeper of mitochondria", the authors studied the formation of VDAC1 clusters in model membranes consisting of mainly PC and PE with varying concentrations of added cholesterol. The authors found that the reconstituted mitochondrial channel VDAC1 forms clusters in the mixed PC/PE bilayer similar to those obtained from native MOM. The clusters mostly vanished at ~2% cholesterol but started to reappear when cholesterol reached above >5%. Overall, I think this work has a lot of potential to be important and insightful if the authors can elaborate on their data processing and statistical practice. Also, it's really a missed opportunity not to look at other lipid species that are either more abundant than cholesterol in MOM (e.g., PIs; see FEBS 1993 330:71-76, BBA 1997 1325:108-116) or those known to cooperate or disrupt cholesterol microdomain (SM/ceramide; see JBC 2004 279:9997-10004, J. lipid res 2020 61:1025-1037) but perhaps these could be reserved for a follow-up study.

I have the following comments on the current study:

- 1) It is unclear whether the microscope images on the same rows of Fig4 are from different experiments or just different zooms of the same sample. It seems there are cases of both. If there is only one experiment per cholesterol %, how can we make sure the 0% cluster is not an anomaly? The scale bars in SFig5 indicate the cluster at 0% cholesterol is particularly small, could that mean VDAC clusters rarely under 0% cholesterol?
- 2) I am not familiar with AFM technology, so I am not sure if it's feasible to have a total count of VDAC particles at different cholesterol %. Assuming we have control of roughly how much VDAC is loaded onto the microscopy sample (deriving from the protein concentration and the liquid volume), having both the % of the loaded protein found in honeycomb clusters and their packing density (RDF) in the clusters should provide a more complete picture than the current manuscript.
- 3) The authors can use more descriptions to elaborate on the algorithm for RDF calculation. Much of the detailed information is missing, so it's difficult to judge the quality of the data. For example, how large is a pixel (resolution) in the raw images analyzed? Is the RDF calculated from any pixel assigned as the protein's density or only from the calculated geometric center of the VDAC circles? Does RDF calculation exclude pixels assigned to the same VDAC particle (self)? It seems to me that the proteins in those false-colored images are not always surrounded by yellow edges, so there might be some algorithm to determine whether a non-protein pixel belongs to a "bound" lipid or an "interstitial" one. Can these two kinds of lipids be distinguished by their heights?
- 4) I would not annotate each column of Fig5 as exactly one extra layer of lipid from another, because PC, PE and cholesterol have very different footprints. On the other hand, I feel SFig7 is more informative than Fig5 -- that VDAC forms a different kind of cluster at 7.5-10% cholesterol (more tightly packed) than at 0% cholesterol because of an early-rising RDF and peaking at a shorter distance.

5) The authors' CG simulation shows the channel prefers to be surrounded by PE than PC, and it seems 50:50 PC/PE yields more clusters than 65:35 PC/PE (SFig2 a,b), plus SFig7 data might indicate a loose cluster under 0% cholesterol whereas a tighter clustering happens above 5% cholesterol. All this information leads to a hypothesis of PE-mediated clustering that could be inhibited by excess PC or cholesterol. The authors might really want to investigate whether the clustering happens under 98:2 and 95:5 POPE:cholesterol (even 100% PE), and contrast that with the results with PC at the same concentrations (SFig2 c,d). If VDAC forms clusters at high PE% without PC, that might explain the disappearance of clusters at 2-5% cholesterol as the result of the competition between two different clustering mechanisms (PE bridged vs cholesterol bridged).

6) What exactly is SM2 in the CG simulations? In page 7 it says POPC/POPE/Chol/cardioliolipin at 62.5/31.5/5.3/1.1 and in Fig6 caption it's POPC/POPE/Chol/CL 58/31/10/1 and in Methods it's 65:35 PC/PE + 10% cholesterol. Since the total lipid count is 1100 molecules, why not list the raw number of each kind?

7) I can't find any information regarding how the simulation data is processed, and the description of simulation protocols is too brief. Can the authors explain the simulation designs in the CHARMM-GUI supplied protocol? For example, was the protein restrained with an elastic network or positional harmonic potential? If it's simple harmonic, do they apply to all protein beads or only on the backbone? How frequently is the trajectory saved for analysis? And finally, how is the CG simulation converted to the density map in Fig6? Is a kernel density function applied to the simulation trajectory from 10 μ s-20 μ s? How large is a voxel in Fig6? What are 1.2x and 1.8x enrichments? Why is more enrichment yielding less density shown in the figure? If PC dominates the space with lower enrichment, doesn't it mean it is the preferred species?

Some minor issues:

a) In Fig6, the sphere of E73 is not easily distinguishable from cholesterol density. Perhaps considering a different color not close to any of the lipids, e.g., green?

b) At 3rd paragraph from bottom page 9: ...honeycombs at PE:PC (65:35) and not (50:50). I'm not sure if it's a typo of PC:PE or it's a different experiment with data not shown.

c) On page 10, the ER and IP3R acronyms were not introduced.

Reviewer #2

(Remarks to the Author)

This manuscript authored by Elodie Lafargue et al. presents an Atomic Force Microscopy study of the ion channels, mimicking in vitro their organization at the mitochondrial membrane. This work is quite complete, including both experiments and molecular dynamics simulation. It describes novel effects of lipids and cholesterol in the architecture of ion channels and deserves publication after attending some minor issues.

1. Lines 174-176 convey the idea that little cholesterol concentration alters the original VDAC organization in linear assemblies. However, adding more cholesterol recovers the original organization. This is explained in the discussion section: "We hypothesize that 2% cholesterol disrupts the PE network at protein contacts without replacing it, causing honeycomb disorganization. As cholesterol content increased, more cholesterol molecules bound around the protein, restoring the honeycomb structure: with 7.5% cholesterol closely resembling the 309 cholesterol-free state." Is it possible to deepen in this explanation? Is there any way to confirm or prove this hypothesis?

2. In Fig. 3b It is difficult to appreciate the existence of aligned molecules in parallel, because choosing the convenient line looks a little arbitrary. Is it possible to add more lines to show this alignment in a clear way?

3. In line 331, the authors discuss the structural rigidity of assemblies. The manuscript reads "All honeycomb assemblies exhibit stability over several minutes.." Why do they become unstable after several minutes? AFM imaging or their very nature? In line 334 the discussion of rigidity results misleading, because something rigid is supposed to be mechanically stiff. What does rigidity mean here?

4. In line 163 the manuscript reads "... is less compact and more dynamic..". I do not see how this dynamic behavior can be derived just looking to Fig. 4a-d.

5. Figure 1 is not properly referred. Line 57 cites Fig. 1d, but there are no mentions to panels a-c before.

6. In line 81, it should read Supplementary figure because it is the beginning of phrase.

Reviewer #3

(Remarks to the Author)

The authors have attempted to explain the influence of lipids on the process of VDAC oligomerization, which is indeed an important aspect to look into in detail. The work carried out by the authors is highly appreciated and needs careful examination, interpretation, and analysis including statistical analysis. I think the authors have not yet explored this in detail about their hypothesis. The hypothesis is not novel to my knowledge, because various research groups have explained the role of lipids in VDAC oligomerization such as in (<https://doi.org/10.1038/s41467-023-43570-y> ; <https://doi.org/10.1016/j.bpj.2011.12.049>).

My concerns are noted below.

Comments:

1. Page 1, Line 19-25 – the hypothesis presented here and the results concluded seem to be not original since various other

groups including Betaneli et al., 2012 and Jahn et al., 2023 have the same conclusion. It is suggested that the authors look back at the data and draw more relevant conclusions which are quite evident from the manuscript like the role of Cholesterol in oligomerization of VDAC.

2. Page 1, Line 29, 30 – molar is incorrect, it is just the percentage.

3. Page 2, Line 64 – the authors do not state which VDAC they are using, Is VDAC or VDAC1? It is confusing for the readers. VDAC1 occurs in multiple locations in the manuscript as well.

4. What is the source of VDAC or in the case VDAC1 (which organism) for the AFM experiments? The authors may demonstrate gel and western blot pictures of the VDAC purified. Even the state of oligomerization can be demonstrated through blotting or other biochemical experiments.

5. Page 6, Line 215 – “induce a curvature the membrane.” The phrase is not clear.

6. Page 6, Line 218-220 – include the explanation for the loss of linearity and other phenomenon in the DISCUSSION.

7. Fig. 5 – What is the explanation for the sudden drops in the bar chart at 5% cholesterol? Why not at another percentage?

8. Page 7, Line 259-261 – Why do the authors call it a “simple membrane”? Why are they not conducting their experiment with a complex membrane which is more realistic?

9. Fig. 6 – “VDAC organizes lipids” What do the authors mean by this statement?

10. Discussion – the discussion of the manuscript is very poor. The authors are advised to review the data and make meaningful conclusions. The interpretations of their results are vastly missing. The authors report their findings again in the discussion which is not appreciated. The authors are advised to discuss their results here and interpret the experimental or computational outcomes. Multiple hypotheses are put forward here based on other findings which is not encouraged. The authors need to solely rely on their results and give future directions, which may be supported by other groups' findings.

11. Since the authors are interested in the oligomerization of VDAC; I encouraged them to estimate the number of clusters or oligomers or simply the oligomeric size of VDAC under various lipid compositions and environments.

12. Cholesterol seems to have meaningful impacts on the formation of clusters which require deep-diving exploration, analysis, and interpretation.

13. The authors may even consider looking at the pore size formed by the oligomeric VDAC which is not discussed here or explored.

14. Since it is a lipid temperature may play an important role which is worth looking at.

15. After all, the title of the manuscript is not convincing. The authors are suggested to change the title after careful reanalysis of their data. Thus, the abstract and the contents of the manuscript including their hypothesis need drastic changes.

Version 1:

Reviewer comments:

Reviewer #1

(Remarks to the Author)

The revised manuscript "Lipids modulate the organization of VDAC1, the gatekeeper of mitochondria" included more in-depth data analysis and the details of these analysis methods, and it's strengthened by additional discussions directly relevant to the experiment data. I really like the updated manuscript and I wish the following minor issues could be addressed when publishing:

a) In the CG simulation protocols, the authors did not mention any positional restraints during the production phase so I assumed the protein structure is solely maintained on elastic networks. The authors only mentioned model building from CHARMM-GUI using MARTINI2.2 with elastic networks. Please elaborate a bit on those parameters (distance cutoffs, force constants, etc.), and it would be best if the authors could provide the RMSD to confirm the protein stays stable in the CG simulation.

b) I still cannot agree with the author's annotation that the 5 Å binning size in fig.5 accounts for one layer of lipid each. We know phospholipids typically occupy 60-70Å² area per molecule (Biochimica et Biophysica Acta 1808: 2761–2771; Biophysical Journal 107: 2274–2286; ACS Omega 4: 10687-10694). Considering that in a hexagonal lattice of densest packed circles, a unit cell area of 60-70Å² corresponds to the circle diameter at 8.3-9Å (area = $\sqrt{3}/2 \times \text{diameter}^2$). In a non-gel phase membrane, the packing of lipids must be less dense than in the hexagonal lattice, i.e., the mean lipid-lipid distance should be greater than 9Å apart, so we might be more realistic to consider 2 bins in fig.5 accounting for one layer of lipid.

c) The average cluster size in table 1 should come with their standard deviations.

d) There is a mixed usage of nm (8 times) and Å (16 times) in the article. Might consider unifying to only one unit in case of confusion.

e) Typo & stylistic inconsistency:

i) 1st paragraph in Results: "betweenn" 33-35

ii) Fig.1 caption: "In AFM imaging, ,"

iii) Page 4, 6His-tag vs 6histag in fig.3 caption

iv) Page 5, (fmono) vs subscript mono in table 1 & caption

v) Page 10, (-0.5<<-5 kJ/mole), I assume the authors meant to say "interactions in the range of -0.5 to -5 kJ/mol"

Reviewer #2

(Remarks to the Author)

Reviewer #3

(Remarks to the Author)

The authors have vastly improved their manuscript and addressed all the concerns raised with more clarity. It may be accepted for publication after some minor changes enumerated below.

1. Table 1: What do the authors mean by “VDAC in structures”?

2. Fig 5 – the author's response to my previous comment read like this - “The integral of the calculation is not constant and depends on the number of protein-protein contacts. Therefore, we do not expect the bars to have uniform heights, as they are influenced by the degree of compaction within the cluster. In the presence of 5% cholesterol, the lipid areas within the clusters are larger compared to the other conditions., which explains the lower bars.”

I am still not fully convinced. I would like the authors to clarify further on this.

I expect the bar chart to continually increase/decrease. However, from the figure, it is noticeable that the bar chart drops at 5% chol compared to no chol, and then at 7.5 and 10% chol; its height increases. What is happening at the %% chol? Are there any possible molecular or chemical changes? Kindly explain.

3. Discussion – I encouraged the authors to avoid repetition of the same sentences or paragraph, unless otherwise necessary. I noticed some of the sentences and paragraphs of DISCUSSION are a repetition of the Results and Introduction section.

4. Table 2 – What is MS1 and MS2?

5. There are some typo errors like 6his-tag. It should be 6His-tag. Kindly correct other errors if any more.

6. Line 586, 587 – “The coarse-grain structure of hVDAC1 was based on that of mouse VDAC1 (PDB ID: 3EMN)37 that 586 contain only 4 conservative mutations over 254 residues.”

This statement is quite confusing. Could the authors clarify this?

Why did the authors choose to model the hVDAC1 from the mVDAC1 PDB structure? My point is hVDAC1 PDB structure is also available in the databases. On the other hand, the AFM study is based on mouse VDAC1, why the author did not choose to continue with mVDAC1

Version 2:

Reviewer comments:

Reviewer #1

(Remarks to the Author)

The authors have fully addressed my concerns listed in the previous reviews.

Reviewer #3

(Remarks to the Author)

The authors have studied the role of lipids in VDAC oligomerization and cluster formation. It is an interesting study to augment the structure-function relationship of VDAC. However, some major concerns still persists despite the authors efforts on drawing the conclusion from their studies.

Major concern:

1. Table 1, Fig 5, SFig 5 and SFig 8 : Should not there be more number of VDAC clusters on adding cholesterol? Should not there be an increase in the degree of compactness with the increase in the percentage of cholesterol? Interpret the outcomes and discuss if any underlying mechanism is involved.

2. Fig 5 and Table 1 : It appears to me that the the VDAC cluster size is much larger with “no chol” as compared to 5% chol. Why and how? Compare with 7.5% and 10% also. Discuss and explain the probable mechanism that might have brought about this inconsistency. The author may consider looking at this article: <https://doi.org/10.1021/acsomega.0c06061> for comparison on the VDAC cluster size.

3. SFig 5 : The author mentioned “VDAC controls the compaction of VDAC cluster”. However, the figure does not demonstrate the above quoted text. 5% chol is supposed to be more compact than “no chol” and the degree of compaction is supposed to increase linearly with the increase in the percentage of cholesterol. Discuss and justify why and how it is taking place.

4. SFig 8 : Similarly, VDAC cluster are more compact with “no chol” as compared to 2% and 5% chol. Why and how? Discuss and explain why there is more number of higher ordered oligomers in 10% chol.

Minor concern:

5. Line 520 : Avanti Polar Lipids

6. Line 526 : Bio-Beads – check it if there are any other errors as well.

7. Line 534, 536 : CaCl₂ (2 in lower superscript) – chect it all.

8. Line 538 : “Nanoscope IIIe Multimode AFM”. Kindly check if it is just “III” or IIIe or something else.

Version 4:

Reviewer comments:

Reviewer #3

(Remarks to the Author)

The author had addressed all my concerns and queries.

Recommend for publication after minor changes. No further review is required.

To avoid confusion among the readers, kindly take note of the point below.

1. As the author have change some terminology, I advised them to be consistent with the terms being used in the entire manuscript. For example: Do the author imply the term "aggregate and cluster" have the same meaning?
2. In my opinion, the change in the percentage of cholesterol might have cause several consequences to the honeycomb structures. Some molecular reasons which I expect might be due to the disruptions in the electrostatics forces interaction or intermolecular interactions. I suggest the author to include such kind of explanations if any. Future experiments can be designed on this line, if not at present study.

We thank the reviewers for their time and there constructive comments.

Reviewer #1 (Remarks to the Author):

In the manuscript "Lipids modulate the organization of VDAC1, the gatekeeper of mitochondria", the authors studied the formation of VDAC1 clusters in model membranes consisting of mainly PC and PE with varying concentrations of added cholesterol. The authors found that the reconstituted mitochondrial channel VDAC1 forms clusters in the mixed PC/PE bilayer similar to those obtained from native MOM. The clusters mostly vanished at ~2% cholesterol but started to reappear when cholesterol reached above >5%. Overall, I think this work has a lot of potential to be important and insightful if the authors can elaborate on their data processing and statistical practice. Also, it's really a missed opportunity not to look at other lipid species that are either more abundant than cholesterol in MOM (e.g., PIs; see FEBS 1993 330:71-76, BBA 1997 1325:108-116) or those known to cooperate or disrupt cholesterol microdomain (SM/ceramide; see JBC 2004 279:9997-10004, J. lipid res 2020 61:1025-1037) but perhaps these could be reserved for a follow-up study.

In this study, we focused on three major lipid components of the mitochondrial outer membrane (MOM): phosphatidylcholine (PC) and phosphatidylethanolamine (PE), the two dominant lipids, along with cholesterol, a critical MOM lipid known to interact with VDAC. While interactions between VDAC1 and PE or cholesterol have been established, their precise roles in VDAC1 organization and clustering remain poorly understood. We clarified that in the text.

In the future, evaluating the effects of other lipids will be crucial for a more comprehensive understanding, and the lipids cited are a key focus of a planned follow-up study.

I have the following comments on the current study:

1) It is unclear whether the microscope images on the same rows of Fig4 are from different experiments or just different zooms of the same sample. It seems there are cases of both. If there is only one experiment per cholesterol %, how can we make sure the 0% cluster is not an anomaly?

Each reconstitution was performed at least three times to ensure the reproducibility of our observations, as clarified in Fig. 4 caption.

A representative subset of images was selected for inclusion in the manuscript. The images selected for figure 4 contain a mixture of zooms from the same image (f), multiple images from the same experiment (a, b, e, f, i) and images from different experiments (c, h, l, j, k, m).

The scale bars in SFig5 indicate the cluster at 0% cholesterol is particularly small, could that mean VDAC clusters rarely under 0% cholesterol?

The reviewer is right, if VDAC clusters at 0% cholesterol, it is forming fewer clusters of smaller assemblies, highlighting the essential role of cholesterol in facilitating the formation

of larger and more organized VDAC assemblies. It is also visible in Fig. 5. We added this to the discussion:

“ Even in the absence of cholesterol, VDAC forms protein-lipid clusters, although these are fewer in number and smaller in size compared to those observed with 5% or more cholesterol.”

2) I am not familiar with AFM technology, so I am not sure if it's feasible to have a total count of VDAC particles at different cholesterol %. Assuming we have control of roughly how much VDAC is loaded onto the microscopy sample (deriving from the protein concentration and the liquid volume), having both the % of the loaded protein found in honeycomb clusters and their packing density (RDF) in the clusters should provide a more complete picture than the current manuscript.

Prior to deposition in the AFM set up, the sample undergoes incubation outside the observation chamber. The adsorption process entails placing the sample, proteoliposomes containing VDAC1, onto a mica surface in the presence of divalent ions. This step allows the formation of a supported lipid bilayer. Following adsorption, the sample is thoroughly rinsed and transferred to the AFM system, leaving no free-floating particles in the solution within the imaging chamber. Adsorption efficiency is governed by a range of factors, including lipid composition, protein concentration, ionic strength, and buffer pH. These variables can significantly affect the uniformity and density of the adsorbed particles, complicating precise enumeration. It is therefore not possible to quantify the total number of particles absorbed on the mica surface.

We added this sentence to Fig. 1 caption for more clarity: “Following adsorption, the sample is thoroughly rinsed and transferred to the AFM system, leaving no free-floating particles in the solution within the imaging chamber.”

3) The authors can use more descriptions to elaborate on the algorithm for RDF calculation. Much of the detailed information is missing, so it's difficult to judge the quality of the data. For example, how large is a pixel (resolution) in the raw images analyzed?

The RDF is now better described in the method section and the code is available.

The pixel resolution is:

CHOL	A/pixel
no CHol	1
2.00%	1.23
5.00%	3.58
7.5%	3.52
10.00%	1.79

Is the RDF calculated from any pixel assigned as the protein's density or only from the calculated geometric center of the VDAC circles? Does RDF calculation exclude pixels assigned to the same VDAC particle (self)?

In AFM imaging, the lateral (XY) resolution is determined by the size of the apex of the AFM tip (extremity of tip), typically ranging from 0.5 to 2 nm, which is sufficient for submolecular imaging. The vertical (Z) resolution is higher, achieving approximately 1 Å, providing exceptional detail in the height dimension. This point has been clarified in caption of Fig. 1.

The radial distribution function (RDF) is calculated based on the geometric centres of the VDAC circles identified in the AFM images, with pixels corresponding to the same VDAC excluded from the analysis. This point has been clarified in the method section.

The Code to calculate the RDF is accessible here:
https://github.com/CodeCanSolveAll/Particle_Analysis.

It seems to me that the proteins in those false-colored images are not always surrounded by yellow edges, so there might be some algorithm to determine whether a non-protein pixel belongs to a "bound" lipid or an "interstitial" one. Can these two kinds of lipids be distinguished by their heights?

We comment to the reviewer that resolving individual lipids between proteins is inherently constrained by the resolution limits of AFM. The topographic resolution is influenced by the convolution of the sample's topography with the shape of the AFM tip apex. When two structures, such as a protein and a closely associated lipid, are in close spatial proximity, this convolution effect reduces the ability of AFM to clearly differentiate them.

4) I would not annotate each column of Fig5 as exactly one extra layer of lipid from another, because PC, PE and cholesterol have very different footprints. On the other hand, I feel SFig7 is more informative than Fig5 -- that VDAC forms a different kind of cluster at 7.5-10% cholesterol (more tightly packed) than at 0% cholesterol because of an early-rising RDF and peaking at a shorter distance.

The reviewer is correct that while PC and PE are structurally similar, the presence of methyl groups on the PC head group significantly influences lipid behavior, as evident from its phase diagram. In the CG model, this difference is accounted for by adjusting the size of the head group bead through modifications to the Lennard-Jones potential. Cholesterol, on the other hand, is structurally distinct. Nevertheless, PC and PE remain the predominant lipid species (90–100%). The representation in Fig. 5 illustrates an approximation of a potential additional lipid layer, as can be visualized in SFig. 3, with exact distances provided in the figure legend. We included this 'lipid layer' annotation for clarity but are open to removing it if deemed unnecessary. Although SFig. 7 provides more detailed information, we feel it is less accessible to a broader audience, which is why we chose to include it in the supplementary material.

5) The authors' CG simulation shows the channel prefers to be surrounded by PE than PC,

and it seems 50:50 PC/PE yields more clusters than 65:35 PC/PE (SFig2 a,b), plus SFig7 data might indicate a loose cluster under 0% cholesterol whereas a tighter clustering happens above 5% cholesterol. All this information leads to a hypothesis of PE-mediated clustering that could be inhibited by excess PC or cholesterol. The authors might really want to investigate whether the clustering happens under 98:2 and 95:5 POPE:cholesterol (even 100% PE), and contrast that with the results with PC at the same concentrations (SFig2 c,d). If VDAC forms clusters at high PE% without PC, that might explain the disappearance of clusters at 2-5% cholesterol as the result of the competition between two different clustering mechanisms (PE bridged vs cholesterol bridged).

Unfortunately, PE is not a lamellar lipid and requires sufficient PC to form stable bilayers. Consequently, we cannot test compositions with 98–95% or 100% POPE, which would, in any case, be non-physiological due to the high PC content of the MOM.

6) What exactly is SM2 in the CG simulations? In page 7 it says POPC/POPE/Chol/cardioliipin at 62.5/31.5/5.3/1.1 and in Fig6 caption it's POPC/POPE/Chol/CL 58/31/10/1 and in Methods it's 65:35 PC/PE + 10% cholesterol. Since the total lipid count is 1100 molecules, why not list the raw number of each kind?

The lipid notation is now homogeneous across all the manuscript. The table 2 has been added to the method section to give the raw composition of the two systems.

7) I can't find any information regarding how the simulation data is processed, and the description of simulation protocols is too brief. Can the authors explain the simulation designs in the CHARMM-GUI supplied protocol? For example, was the protein restrained with an elastic network or positional harmonic potential? If it's simple harmonic, do they apply to all protein beads or only on the backbone? How frequently is the trajectory saved for analysis? And finally, how is the CG simulation converted to the density map in Fig6? Is a kernel density function applied to the simulation trajectory from 10 μ s-20 μ s? How large is a voxel in Fig6? What are 1.2x and 1.8x enrichments? Why is more enrichment yielding less density shown in the figure? If PC dominates the space with lower enrichment, doesn't it mean it is the preferred species?

A new paragraph in the methods section explicitly gives the details of the CHARMM-GUI supplied protocol, including the information on the step by step decrease of the harmonic position restraints on both protein backbone and lipid head groups.

The legend for Figure 6 has been updated to clarify the interpretation of lipid density enrichment relative to the average density. Additionally, a new paragraph has been added to the Materials and Methods section, providing detailed information about the procedure used to construct the volumetric map and explaining how it aids in addressing convergence among the replicas.

Some minor issues:

a) In Fig6, the sphere of E73 is not easily distinguishable from cholesterol density. Perhaps considering a different color not close to any of the lipids, e.g., green?

We changed the sphere's color to black for better visualization.

b) At 3rd paragraph from bottom page 9: ...honeycombs at PE:PC (65:35) and not (50:50). I'm not sure if it's a typo of PC:PE or it's a different experiment with data not shown.

It was a typo, thank you, we corrected it.

c) On page 10, the ER and IP3R acronyms were not introduced.

We added them.

Reviewer #2 (Remarks to the Author):

This manuscript authored by Elodie Lafargue et al. presents an Atomic Force Microscopy study of the ion channels, mimicking in vitro their organization at the mitochondrial membrane. This work is quite complete, including both experiments and molecular dynamics simulation. It describes novel effects of lipids and cholesterol in the architecture of ion channels and deserves publication after attending some minor issues.

1. Lines 174-176 convey the idea that little cholesterol concentration alters the original VDAC organization in linear assemblies. However, adding more cholesterol recovers the original organization. This is explained in the discussion section: "We hypothesize that 2% cholesterol disrupts the PE network at protein contacts without replacing it, causing honeycomb disorganization. As cholesterol content increased, more cholesterol molecules bound around the protein, restoring the honeycomb structure: with 7.5% cholesterol closely resembling the 309 cholesterol-free state.." Is it possible to deepen in this explanation? Is there any way to confirm or prove this hypothesis?

We expanded on this explanation in the discussion. However, experimentally confirming the hypothesis would be challenging, as it lies at the current limits of our capabilities. Investigating the precise interactions between a protein and specific lipids could be accomplished using masNMR, a technique that is not easily accessible and requires large quantities of NMR-labeled proteins and lipids. Alternatively, photolabeling could provide more precise and quantitative results, but it is also difficult to implement and less accessible without photo-activable lipids.

2. In Fig. 3b It is difficult to appreciate the existence of aligned molecules in parallel, because choosing the convenient line looks a little arbitrary. Is it possible to add more lines to show this alignment in a clear way?

We updated the figure for more clarity.

3. In line 331, the authors discuss the structural rigidity of assemblies. The manuscript reads “All honeycomb assemblies exhibit stability over several minutes..” Why do they become unstable after several minutes? AFM imaging or their very nature? In line 334 the discussion of rigidity results misleading, because something rigid is supposed to be mechanically stiff. What does rigidity mean here?

The honeycomb structures do not become unstable after a few minutes; they remain stable throughout the imaging sequence within the experimental timeframe. VDAC assemblies are not crystalline but exhibit "glass-like" behavior, indicating that while they maintain structural stability, they are not frozen and can reorganize their components in response to environmental conditions, such as local lipid variations or mechanical perturbations. These assemblies preserve their structures despite minor influences, such as tip-induced forces, temperature fluctuations, or changes in lipid composition. Lipids play a crucial role in stabilizing the proteins within the glass-like network, functioning as a "molecular mortar" that supports the cohesion and adaptability of the clusters, as we detailed in the discussion section.

We modified the discussion and now mention stability and dynamic rather than rigidity, which was indeed confusing.

4. In line 163 the manuscript reads “... is less compact and more dynamic..”. I do not see how this dynamic behavior can be derived just looking to Fig. 4a-d.

We have detailed this information from sequences of AFM images. Additionally, a new supplementary movie (movie 1), composed of a sequence of AFM images, has been added to illustrate the dynamic nature of VDAC assemblies in a lipid composition of PC:PE:2%Chol.

5. Figure 1 is not properly referred. Line 57 cites Fig. 1d, but there are no mentions to panels a-c before.

It was corrected.

6. In line 81, it should read Supplementary figure because it is the beginning of phrase.

It was corrected.

Reviewer #3 (Remarks to the Author):

The authors have attempted to explain the influence of lipids on the process of VDAC oligomerization, which is indeed an important aspect to look into in detail. The work carried out by the authors is highly appreciated and needs careful examination, interpretation, and analysis including statistical analysis.

As with any imaging technique, and perhaps even more so with AFM, drawing statistical conclusions can be challenging. First, the analysis is limited to what is observed, and in AFM, this involves several selections: what adheres to the support (which the experimenter has little control over), what is observable (sufficiently slow-moving to be imaged), and what the observer chooses to focus on. Consequently, obtaining reliable statistical data on many aspects is difficult. In contrast, molecular dynamics simulations offer a more robust approach for statistical analysis, allowing for the examination of convergence and variability across multiple simulations.

I think the authors have not yet explored this in detail about their hypothesis. The hypothesis is not novel to my knowledge, because various research groups have explained the role of lipids in VDAC oligomerization such as in (<https://doi.org/10.1038/s41467-023-43570-y> ; <https://doi.org/10.1016/j.bpj.2011.12.049>).

While the hypothesis is not novel for proteins in general, it has been minimally explored for VDAC, particularly at the structural level. We acknowledge that it was an oversight not to cite the work by Betaneli et al., and this citation has been added to the manuscript. However, our study differs significantly from theirs, as we focus on different lipids. Furthermore, we extend their findings by highlighting the importance of cholesterol and providing nanoscale visualization of the size, compaction, and stability of lipid clusters, associated to molecular dynamic simulations.

In contrast, the study by Jhan et al. does not address the role of lipids in VDAC oligomerization but instead examines the function of VDAC as a lipid scramblase. Their work demonstrates the role of dimers in lipid scrambling but does not investigate the influence of specific lipids on this activity. We have appropriately cited both studies in our manuscript.

My concerns are noted below.

Comments:

1. Page 1, Line 19-25 – the hypothesis presented here and the results concluded seem to be not original since various other groups including Betaneli et al., 2012 and Jahn et al., 2023 have the same conclusion. It is suggested that the authors look back at the data and draw more relevant conclusions which are quite evident from the manuscript like the role of Cholesterol in oligomerization of VDAC.

We have revised our introduction to include a citation to the study by Betaneli et al. Our study significantly extends that of *Betaneli et al.* Unlike their work, we emphasize the role of cholesterol and demonstrate the nanoscale properties of VDAC clusters, including their size, compaction, and stability.

While the study by Jhan et al. is already cited in our paper, it does not address the effect of lipids on VDAC oligomerization. Additionally, we have updated our conclusion to place

greater emphasis on the role of cholesterol in VDAC oligomerization, however, our study show an important role for PE as well, and not only cholesterol.

2. Page 1, Line 29, 30 – molar is incorrect, it is just the percentage.

We removed “molar”.

3. Page 2, Line 64 – the authors do not state which VDAC they are using, Is VDAC or VDAC1? It is confusing for the readers. VADC1 occurs in multiple locations in the manuscript as well.

We consistently use VDAC1 throughout the study, as detailed in the Methods section. Specifically, we utilize mouse VDAC1, which is expressed and purified from *E. coli* and subsequently reconstituted into liposomes. This clarification has been added to the manuscript.

4. What is the source of VDAC or in the case VDAC1 (which organism) for the AFM experiments? The authors may demonstrate gel and western blot pictures of the VDAC purified. Even the state of oligomerization can be demonstrated through blotting or other biochemical experiments.

We have added a supplementary figure (SFig. 10) showing our purified mouse VDAC1 in its monomeric form in detergent (in LDAO detergent, VDAC1 remains monomeric). The protocol is described in great details in the reference Dearden et al. (Star Protocol), cited in our manuscript.

In this study, we investigate the impact of lipids on VDAC1 oligomerization, utilizing AFM to directly visualize the protein within the membrane—a highly convincing method for this purpose. To our knowledge, no method currently exists for performing Western blotting in lipid membranes.

5. Page 6, Line 215 – “induce a curvature the membrane.” The phrase is not clear.

We removed it.

6. Page 6, Line 218-220 – include the explanation for the loss of linearity and other phenomenon in the DISCUSSION.

We change “linearity” for filament-like, and discussed this phenomenon in the discussion.

7. Fig. 5 – What is the explanation for the sudden drops in the bar chart at 5% cholesterol? Why not at another percentage?

The integral of the calculation is not constant and depends on the number of protein-protein contacts. Therefore, we do not expect the bars to have uniform heights, as they are influenced by the degree of compaction within the cluster. In the presence of 5% cholesterol, the lipid areas within the clusters are larger compared to the other conditions., which explain the lower bars.

We added to the text:

“Figure 5 illustrates the VDAC-VDAC distances within the clusters under varying cholesterol conditions (from 0 to 10%). For each cholesterol concentration, we categorized interactions into three types: direct protein-protein interactions (<46 Å), lipid-bridged interactions, and mixed lipid-protein-bridged interactions. As shown in the AFM data, there are fewer protein-protein contacts at 5% cholesterol, likely due to a less compact assembly, which corresponds to the marked decrease in the bar chart at this concentration. Conversely, as cholesterol levels increase, the clusters become more compact, as evidenced by the overall rise in the bar chart, especially in the direct protein-protein interactions (first bar), reflecting enhanced compaction.”

8. Page 7, Line 259-261 – Why do the authors call it a “simple membrane”? Why are they not conducting their experiment with a complex membrane which is more realistic?

Our simple membrane (SM) refers to a model membrane that approximates the composition of the mitochondrial outer membrane (MOM) using only two (SM1) or three (SM2) lipids: POPC, POPE, and cholesterol. We clarified it in the manuscript.

Studying the effects of specific lipids in a more complex membrane environment poses significant challenges, as the relative abundance of individual lipids is much lower, leading to reduced convergence in molecular dynamics (MD) simulations. Additionally, AFM experiments in these environments are time-consuming and can be difficult to interpret due to the increased complexity. Despite these challenges, we plan to explore a more complex membrane environment in a follow-up study to better mimic physiological conditions and lipid interactions.

9. Fig. 6 – “VDAC organizes lipids” What do the authors mean by this statement?

We mean that the distribution of lipids around the protein is not random.

We modified the title of Fig. 6 to clarify this statement, and clarified the text.

“The molecular dynamics simulation reveals that VDAC influences the local lipid distribution within the membrane, creating distinct lipid layers. Cholesterol (Chol) and phosphatidylethanolamine (PE) are preferentially located in the first lipid layer in direct contact with VDAC, while phosphatidylcholine (PC) is more prevalent in a second, adjacent layer.”

10. Discussion – the discussion of the manuscript is very poor. The authors are advised to review the data and make meaningful conclusions. The interpretations of their results are vastly missing. The authors report their findings again in the discussion which is not appreciated. The authors are advised to discuss their results here and interpret the experimental or computational outcomes. Multiple hypotheses are put forward here based on other findings which is not encouraged. The authors need to solely rely on their results and give future directions, which may be supported by other groups' findings.

We significantly revised the discussion in response to the reviewer's comments.

11. Since the authors are interested in the oligomerization of VDAC; I encouraged them to estimate the number of clusters or oligomers or simply the oligomeric size of VDAC under various lipid compositions and environments.

We recognize that understanding the impact of lipids on VDAC oligomerization necessitates a thorough quantification of the various types of oligomers present in the proteolipid assemblies we are studying.

To achieve this, we have developed a tool that analyzes these assemblies in terms of both cluster number and size. The implementation of this cluster analysis program is detailed in a newly added section of the Materials and Methods. Additionally, the Results section includes a new table (table 1) and a supplementary figure (SFig. 8), and new paragraphs (pages 5 and 6), that illustrate the evolution of these clusters across different environments.

Among our new findings, we successfully identified and quantified the number of proteins that remain stable while not being in direct contact with other proteins. This exemplifies the previously unanticipated behavior of our minimal outer mitochondrial membrane in facilitating long-range protein interactions, similar to the role of mortar in construction or the characteristics of a glassy environment.

12. Cholesterol seems to have meaningful impacts on the formation of clusters which require deep-diving exploration, analysis, and interpretation.

Indeed, this is an important aspect, and we tried to discuss it more thoroughly in the discussion.

However, for a deeper-diving exploration, coarse-grained (CG) simulations may not be the most suitable tool due to the approximations and limitations in the representation of proteins and lipids, which restrict a detailed analysis. There is currently insufficient data to explore this in depth. Hence, the suggestion of the reviewer falls outside the scope of the present study, but it is a direction we plan to explore in the future.

13. The authors may even consider looking at the pore size formed by the oligomeric VDAC which is not discussed here or explored.

The pore size is the size of the individual VDAC pore, as seen in Fig 1.d, of about 4 nm, corresponding to the solved 3D structure. The lateral resolution of AFM (0.5-1 nm) doesn't allow to visualize small variations of barrel size.

14. Since it is a lipid temperature may play an important role which is worth looking at.

We decided not to investigate this aspect for technical reason, as it is difficult to do a precise temperature titration in AFM

All our experiments were performed at temperatures between 33-35°C to ensure that phase transitions are avoided. At 25°C, the membrane exhibits phase separation (SFig. 1), which is not physiological and could complicate data interpretation significantly.

15. After all, the title of the manuscript is not convincing. The authors are suggested to change the title after careful reanalysis of their data. Thus, the abstract and the contents of the manuscript including their hypothesis need drastic changes.

We vastly modified the discussion and abstract.

We understand and appreciate the reviewer's suggestion to emphasize the role of cholesterol in the title. However, after careful consideration, we believe that changing the title could be potentially misleading for the audience. While our study examines the effects of three lipids (PE, PC, and cholesterol), we think the observed effects are likely more general in nature.

The reference by Betaneli et al., as suggested by the reviewer, highlights the effects of two other lipids (PC and CL) on VDAC, and it is likely that additional lipids may also influence VDAC behavior. Furthermore, other studies focusing on the relationship between VDAC and lipids tend to use titles such as "*The Role of Lipids in VDAC Oligomerization*" (reviewer's suggested reference), "*Membrane Lipid Composition Regulates Tubulin Interaction with Mitochondrial Voltage-Dependent Anion*," and "*Voltage Gating of VDAC is Regulated by Nonlamellar Lipids of Mitochondrial Membranes*," among others, in the same line as our title. For these reasons, we would prefer to retain our original title.

We have modified the abstract to highlight the roles of PE and cholesterol and made modifications to the manuscript to address these key points. We hope this provides sufficient context and clarity.

We thank the reviewers for their comments. Here are our answers:

Reviewer 1:

a) In the CG simulation protocols, the authors did not mention any positional restraints during the production phase so I assumed the protein structure is solely maintained on elastic networks. The authors only mentioned model building from CHARMM-GUI using MARTINI2.2 with elastic networks. Please elaborate a bit on those parameters (distance cutoffs, force constants, etc.), and it would be best if the authors could provide the RMSD to confirm the protein stays stable in the CG simulation.

The "all-atom" VDAC1 structure was coarse-grained using Martinize 2.5 and the MARTINI 2.2 force field, with the Elnedyn elastic network model. This was implemented using an elastic force constant of 500 kJ/mol/nm² between all backbone particles falling below 0.9 nm distance, excluding those pertaining to successive beads. This setup allowed for the production of stable VDAC conformations throughout the 20 μs simulation, as evidenced by the RMSD time series shown in the supplementary material. We inserted this into the methods.

A Sfig. 13 Providing the RMSD was added.

Sfig. 13: Root Mean Square Deviation (RMSD) calculated along the 20 μs trajectories, using the backbone particles of the first production configuration as a reference. Panels A-C correspond to the replicas of hVDAC 1 in Simple Membrane 1, while panels D-F represent the replicas in Simple Membrane 2.

b) I still cannot agree with the author's annotation that the 5 Å binning size in fig.5 accounts for one layer of lipid each. We know phospholipids typically occupy 60-70Å² area per molecule (Biochimica et Biophysica Acta 1808: 2761–2771; Biophysical Journal 107: 2274–2286; ACS Omega 4: 10687-10694). Considering that in a hexagonal lattice of densest packed circles, a unit cell area of 60-70Å² corresponds to the circle diameter at 8.3-9Å (area = $\sqrt{3}/2 \times \text{diameter}^2$). In a non-gel phase membrane, the packing of lipids must be less dense than in the hexagonal lattice, i.e., the mean lipid-lipid distance should be greater than 9Å apart, so we might be more realistic to consider 2 bins in fig.5 accounting for one layer of lipid.

We understand the referee's hesitation regarding our figure, and we acknowledge that the term 'layer' in the figure or 'shell' in the figure legend can be quite misleading.

The differing interpretations stem from our definition of the lipid number from the protein surface, as illustrated in SFigures 3c and 3d. In this perspective, a “layer” may be formed from distinct planes, where a lipid can contribute only a partial moiety, mechanically rising the

Hence, we removed the mention of layers in the text and modified the Fig. 5, results and methods in a way we hope is an improvement.

“Distances between the proteins illustrate the effect of cholesterol on compaction

Figure 5 illustrates the VDAC-VDAC distances **distribution** within the aggregates under varying cholesterol (from 0 to 10%). For each cholesterol concentration, we categorized pairwise neighbor interactions into three types: direct protein-protein interactions ($<53 \text{ \AA}$), lipid-bridged interactions, and mixed lipid-protein-bridged interactions. As is clear in the AFM images, there are many fewer protein-protein contacts at 5% cholesterol (Fig. 4 e-g), which corresponds to the marked decrease in the red bar height in Fig. 5 at this concentration. Conversely, as cholesterol levels increase, the clusters become more compact (Fig. 4 h-o), as evidenced by the overall rise in the bar heights in Fig. 5, especially in the direct protein-protein interactions (first **two** bars), reflecting enhanced compaction. **As the compaction increases, the population of lipid-separated proteins is maintained or even reinforced (as shown in the next two bars).** This suggests that multilayers of lipids persist between a significant fraction of the proteins, even in the dense regime with 10% cholesterol content. To further explore the heterogeneity of protein separation within the aggregates, we analyzed the distance distribution within smaller aggregates, specifically focusing on regions surrounding randomly selected proteins in the structures. The averages obtained from these ensembles (represented by the blue bars) are relatively close to those measured for the entire image. However, the standard deviation, along with the individual measurements (100, though 87 for the smallest structure), reveals considerable variation, indicating substantial heterogeneity within the assemblies.

Supplementary Figure 5 presents the models at higher magnification to enhance the visualization of lipid roles. Discrete peaks observed in the Radial Distribution Functions (SFig. 7) indicate peaks separated by 6 \AA , which represent distinct lipid arrangements in lipid-bridged protein separations, as illustrated in Supplementary Figure 3.

“

c) The average cluster size in table 1 should come with their standard deviations.

The values presented in Table 1 are derived from the analysis of the AFM micrograph using our clustering algorithm. This table represents a quantification of the features observed in a single image. However, the reviewer's concerns are valid, as the results may indeed depend on the specifics of the primary parameter of the algorithm (ϵ), which defines the approach distance for determining contacts.

We have previously selected this value as optimal for accurately reproducing the visual assessment of cluster formation. To evaluate the robustness of this choice and assess the sensitivity of the results to this parameter, we varied the eps value around the original setting to examine its impact on the average cluster size. The results of this analysis, including the new average and standard deviation, are presented in the updated Table 1.

	no Chol	2%	5%	7.5%	10%
# VDAC in structures	87	63	252	330	399
f_{mono}	0.45	0.56	0.98	0.31	0.05
# Clusters	16	10	2	65	14
f_{clust}	0.55	0.44	0.02	0.69	0.95
<Clust size> +/- std dev	3.08+/-0.17	2.70+/-0.09	2.00+/-0.00	3.54+/-0.16	38.38+/-27.59

Table 1 : Distribution of VDAC within the structures, as free monomer or among clusters. # VDAC represents the number of proteins identified in our reference AFM images for the various structures depicted in Figure 2.a and 4.c, f, j m. f_{mono} denote the fraction of VDAC molecules categorized as isolated particles using our DBSCAN algorithm (see Methods). # Clusters indicates the total number of clusters identified. f_{clust} denotes the fraction of VDAC molecules within those clusters. Finally, <Cluster Size> refers to the average number of VDAC calculated across all clusters (excluding isolated monomers). The average and standard deviation of cluster size were obtained by varying the single parameter of the algorithm, specifically the distance between monomers used to determine whether they are interacting.

d) There is a mixed usage of nm (8 times) and Å (16 times) in the article. Might consider unifying to only one unit in case of confusion.

It is done.

e) Typo & stylistic inconsistency:

i) 1st paragraph in Results: "betwenn" 33-35

ii) Fig.1 caption: "In AFM imaging, ,"

iii) Page 4, 6His-tag vs 6histag in fig.3 caption

iv) Page 5, (fmono) vs subscript mono in table 1 & caption

v) Page 10, (-0.5<<-5 kJ/mole), I assume the authors meant to say "interactions in the range of -0.5 to -5 kJ/mol"

We modified all those typos in the text.

Reviewer 3:

1. Table 1: What do the authors mean by “VDAC in structures”?

It is the number of VDAC molecules in the analyzed structures, which are separated into monomers and clusters of different sizes, as explained in the legend.

2. Fig 5 – the author's response to my previous comment read like this - “The integral of the calculation is not constant and depends on the number of protein-protein contacts. Therefore, we do not expect the bars to have uniform heights, as they are influenced by the degree of compaction within the cluster. In the presence of 5% cholesterol, the lipid areas within the clusters are larger compared to the other conditions., which explains the lower bars.”

I am still not fully convinced. I would like the authors to clarify further on this.

I expect the bar chart to continually increase/decrease. However, from the figure, it is noticeable that the bar chart drops at 5% chol compared to no chol, and then at 7.5 and 10% chol; its height increases. What is happening at the %% chol? Are there any possible molecular or chemical changes? Kindly explain.

For instance, it is straightforward to see that when the proteins are dispersed, the number of observation considering contacting residues strongly decreases. As a result, both the number of observations and their distribution are heavily influenced by the specific details of the aggregate structure.

We believe that replacing the error bars with the observed data points for each individual bars will help clarify the situation regarding the number of observations and the distribution of the points.

2. Discussion – I encouraged the authors to avoid repetition of the same sentences or paragraph, unless otherwise necessary. I noticed some of the sentences and paragraphs of DISCUSSION are a repetition of the Results and Introduction section.

We did our best to have the clearer discussion without impairing the understanding. We tried to modify further the discussion to limit repetition.

4. Table 2 – What is MS1 and MS2?
SM1 and SM2, corrected

5. There are some typo errors like 6his-tag. It should be 6His-tag. Kindly correct other errors if any more.

We corrected and checked the document for other typos

6. Line 586, 587 – “The coarse-grain structure of hVDAC1 was based on that of mouse VDAC1 (PDB ID: 3EMN)³⁷ that 586 contain only 4 conservative mutations over 254 residues.”

This statement is quite confusing. Could the authors clarify this?

Why did the authors choose to model the hVDAC1 from the mVDAC1 PDB structure? My point is hVDAC1 PDB structure is also available in the databases.

On the other hand, the AFM study is based on mouse VDAC1, why the author did not choose to continue with mVDAC1

The molecular dynamics study was initially integrated with other ongoing studies on hVDAC1. However, in the light of our AFM observations and the minimal differences between the 2 proteins, we chose to leverage our MD simulations to give a clearer understanding of the role of lipids in VDAC organization.

The 3EMN structure of mVDAC1 was at higher resolution than the hVDAC1 structures available in the databases at the time we started this project.

We clarified this:

“The coarse-grained structure of hVDAC1 was modeled based on the mouse VDAC1 structure (at higher resolution than hVDAC1, PDB ID: 3EMN), which differs by only four conservative

mutations

over

254

residues.

“

We thanks the reviewers for their comments.

Reviewer #1 (Remarks to the Author):

The authors have fully addressed my concerns listed in the preivous reviews.

Reviewer #3 (Remarks to the Author):

The authors have studied the role of lipids in VDAC oligomerization and cluster formation. It is an interesting study to augment the structure-function relationship of VDAC. However, some major concerns still persists despite the authors efforts on drawing the conclusion from their studies.

Major concern:

1. Table 1, Fig 5, SFig 5 and SFig 8 : Should not there be more number of VDAC clusters on adding cholesterol? Should not there be an increase in the degree of compactness with the increase in the percentage of cholesterol? Interpret the outcomes and discuss if any underlying mechanism is involved.
2. Fig 5 and Table 1 : It appears to me that the the VDAC cluster size is much larger with “no chol” as compared to 5% chol. Why and how? Compare with 7.5% and 10% also. Discuss and explain the probable mechanism that might have brought about this inconsistency. The author may consider looking at this article: <https://doi.org/10.1021/acsomega.0c06061> for comparison on the VDAC cluster size.
3. SFig 5 : The author mentioned “VDAC controls the compaction of VDAC cluster”. However, the figure does not demonstrate the above quoted text. 5% chol is supposed to be more compact than “no chol” and the degree of compaction is supposed to increase linearly with the increase in the percentage of cholesterol. Discuss and justify why and how it is taking place.

We changed the title of the figure

4. SFig 8 : Similarly, VDAC cluster are more compact with “no chol” as compared to 2% and 5% chol. Why and how? Discuss and explain why there is more number of higher ordered oligomers in 10% chol.

All these concerns stem from a misunderstanding of the terminology we used. Thanks to the reviewer's comments, we realized our nomenclature was unclear. We have now added explanations (quoted below)at the beginning of the results section and revised the terminology throughout the manuscript to improve clarity. We hope this resolves the issue.

« In the following sections, we will refer to "structures" as protein assemblies that are stable enough to produce wide-field AFM images, regardless of whether they are aggregates of pure proteins or contain lipids. In contrast, "clusters" will refer to oligomers within these structures that show direct protein-protein contacts. »

We thank the reviewer for raising the points which drew our attention to this difficulty of terminology and was exacerbated by the description of the new analyses performed after the first round of reviews. We have now clarified this distinction throughout the text, ensuring

consistency in our use of *oligomerization* and *clusters*, with all occurrences carefully revised in the manuscript.

Regarding the cited paper, while it is indeed interesting, it focuses on a macroscopic study of VDAC function rather than structural aspects, making it fundamentally different from our work.

Minor concern:

5. Line 520 : Avanti Polar Lipids

6. Line 526 : Bio-Beads – check it if there are any other errors as well.

7. Line 534, 536 : CaCl₂ (2 in lower superscript) – check it all.

8. Line 538 : “Nanoscope IIIe Multimode AFM”. Kindly check if it is just “III” or IIIe or something else.

Thanks for these comments we have reread the manuscript with care to verify these points.

We thank the reviewers for their insightful comments, which highlighted the need for greater precision in our terminology. We acknowledge that terms such as 'clusters' and 'assemblies' were previously used inconsistently, leading to ambiguity. In response, we have carefully revised the manuscript to ensure clarity and consistency in the use of these terms throughout

Reviewer #1 (Remarks to the Author):

The authors have fully addressed my concerns listed in the previous reviews.

Reviewer #3 (Remarks to the Author):

The authors have studied the role of lipids in VDAC oligomerization and cluster formation. It is an interesting study to augment the structure-function relationship of VDAC. However, some major concerns still persists despite the authors efforts on drawing the conclusion from their studies.

Major concern:

1. Table 1, Fig 5, SFig 5 and SFig 8 : Should not there be more number of VDAC clusters on adding cholesterol? Should not there be an increase in the degree of compactness with the increase in the percentage of cholesterol? Interpret the outcomes and discuss if any underlying mechanism is involved.

We believe that this concern, along with the subsequent points, stems from two misleading factors: 1) Following the initial round of reviews, we inadvertently created confusion by using the term 'cluster' in various unrelated contexts, and 2) our figure descriptions did not sufficiently highlight the reversible effects of cholesterol on both assembly compaction and cluster formation.

To describe the various modalities of protein assemblies within the membrane, we define key terms as follows:

Clusters: Groups of proteins connected by direct protein-protein interactions within a distance of 5.6 nanometers.

Assemblies: Larger groupings of proteins, which may include both direct protein-protein interactions and lipid-mediated associations.

Structures: Well-defined organizations, as directly observed in AFM micrographs, encompassing the diverse organizational states described above.

Throughout the manuscript, we have carefully applied these definitions to ensure precise and consistent use of terminology when describing the complexity of protein and lipid organization in membranes.

The figures mentioned (Table 1, Fig 5, SFig 5 and SFig 8) do not rely on the same type of information. Table 1 and SFig 8 effectively illustrate the evolution of clusters within various assemblies, while Fig. 5 and SFig 5 focus on the compaction of VDAC within those assemblies. As stated in the preliminary section, we have corrected this misleading information by replacing 'cluster' with 'assembly' where necessary throughout the paper, particularly in the last two figures.

Further details regarding additional changes related to these concerns are discussed more specifically in the following paragraphs.

2. Fig 5 and Table 1 : It appears to me that the the VDAC **cluster** size is much larger with “no chol” as compared to 5% chol. Why and how? Compare with 7.5% and 10% also. Discuss and explain the probable mechanism that might have brought about this inconsistency. The author may consider looking at this article: <https://doi.org/10.1021/acsomega.0c06061> for comparison on the VDAC **cluster size**.

The larger cluster size (Table 1) and compaction (Fig. 5) observed in the 'no cholesterol' condition are very similar to those in the 7.5% cholesterol condition. In contrast, the 5% cholesterol concentration exhibits significantly lower compaction and a reduced number of clusters. This highlights the reversible effect of cholesterol addition on both properties, which we now emphasize throughout the paper.

For Figure 5, we have changed the title to eliminate ambiguity regarding the effect of cholesterol on compaction and the properties we are analysing. The new title, “Reversible Cholesterol-Driven Compaction of VDAC1 Assemblies”, replaces the previous title, 'Increased Cholesterol Triggers Effect on the Compaction of VDAC1 Clusters.' This modification explicitly highlights the initial effect of the increase in cholesterol on protein decompaction, followed by the subsequent restoration of compaction at 7.5% and 10% cholesterol.

To further emphasize the reversibility of the (de)compaction process, we provide additional details in the corresponding results section. Also, the first section of the discussion provides the analysis of these facts.

Figure 5 illustrates the VDAC-VDAC distances distribution within the aggregates under varying cholesterol (from 0 to 10%). For each cholesterol concentration, we categorized pairwise neighbor interactions into three types: direct protein-protein interactions (<53 Å), lipid-bridged interactions, and mixed lipid-protein-bridged interactions. As is clear in the AFM images, there are many fewer protein-protein contacts at 5% cholesterol (Fig. 4 e-g) compared to the cholesterol-free sample, which corresponds to the marked decrease in the ~~red~~ bar heights ~~in~~ (Fig. 5) at between the two this concentrations, and a decrease in the compaction. Conversely, as cholesterol levels further increase, the processus reverse and the clusters assemblies become more compact (Fig. 4 h-o), as evidenced by the overall rise in the bar heights in Fig. 5, especially in the direct protein-protein interactions (first two bars), reflecting enhanced compaction. As the compaction increases, the population of lipid-separated proteins is main-

Regarding Table 1 and the apparent inconsistency in cluster size evolution, we acknowledge the reviewer’s observation about the reversible effect on VDAC cluster growth. This trend

mirrors the previously described biphasic behavior: from 0% to 5% cholesterol, the initial 'honeycomb-like' clusters diminish, leading to near-complete cluster loss at 5%, followed by renewed cluster growth from 5% to 10%. This pattern is analyzed in the first paragraph of the Discussion section

The cited paper is indeed exciting for its exploration of the relationships between a Zimm–Bragg model of protein clustering and the macroscopic study of VDAC conductance; however, it does not directly address the structural aspects of cluster formation. While relating our findings to such a model could offer additional perspectives on our work, we believe it is somewhat beyond the scope of the current paper.

3. SFig 5 : The author mentioned “VDAC controls the compaction of VDAC cluster”. However, the figure does not demonstrate the above quoted text. **5% chol is supposed to be more compact than “no chol”** and the degree of compaction is supposed to increase linearly with the increase in the percentage of cholesterol. Discuss and justify why and how it is taking place.

As previously mentioned, the fact that the 5% cholesterol assembly is indeed less compact than the 'no cholesterol' condition is a consequence of the aforementioned reversible effect of cholesterol at low concentrations. This is now presented without any ambiguity. Also, the misleading title '*Cholesterol controls the compaction of VDAC clusters*' was changed to “**Cholesterol controls the compaction of VDAC assemblies**”.

4. SFig 8 : Similarly, VDAC clusters are more compact with “no chol” as compared to 2% and 5% chol. Why and how? Discuss and explain why there is more number **of higher ordered oligomers in 10% chol**.

Also, the emphasis on the reversible effect of cholesterol on the compactness and the formation of VDAC clusters should alleviate this concern.

At 10% chol the number of proteins within clusters continue to increase relatively to the previous cholesterol concentration however most of them merge into a single large cluster, an oligomer, depicted in blue on SFig 8.

We believe these clarifications strengthen the first paragraph of the Discussion, where we propose a mechanistic explanation for the observed results. Briefly, this mechanism involves composition-dependent, specific or nonspecific interactions between the protein surface and surrounding lipids.

Minor concern:

5. Line 520 : Avanti Polar Lipids

6. Line 526 : Bio-Beads – check it if there are any other errors as well.

7. Line 534, 536 : CaCl₂ (2 in lower superscript) – check it all.

8. Line 538 : “Nanoscope IIIe Multimode AFM”. Kindly check if it is just “III” or IIIe or something else.

Thanks for these comments we have reread the manuscript with care to verify these points.

This manuscript authored by Elodie Lafargue et al. presents an Atomic Force Microscopy study of the ion channels, mimicking in vitro their organization at the mitochondrial membrane. This work is quite complete, including both experiments and molecular dynamics simulation. It describes novel effects of lipids and cholesterol in the architecture of ion channels and deserves publication after attending some issues.

1. Lines 174-176 convey the idea that little cholesterol concentration alters the original VDAC organization in linear assemblies. However, adding more cholesterol recovers the original organization. This is explained in the discussion section: “*We hypothesize that 2% cholesterol disrupts the PE network at protein contacts without replacing it, causing honeycomb disorganization. As cholesterol content increased, more cholesterol molecules bound around the protein, restoring the honeycomb structure: with 7.5% cholesterol closely resembling the 309 cholesterol-free state.*” Is it possible to deepen in this explanation? Is there any way to confirm or prove this hypothesis?
2. In Fig. 3b It is difficult to appreciate the existence of aligned molecules in parallel, because choosing the convenient line looks a little arbitrary. Is it possible to add more lines to show this alignment in a clear way?
3. In line 331, the authors discuss the structural rigidity of assemblies. The manuscript reads “*All honeycomb assemblies exhibit stability over several minutes.*” Why do they become unstable after several minutes? AFM imaging or their very nature? In line 334 the discussion of rigidity results misleading, because something rigid is supposed to be mechanically stiff. What does rigidity mean here?
4. In line 163 the manuscript reads “*... is less compact and more dynamic.*”. I do not see how this dynamic behavior can be derived just looking to Fig. 4a-d.
5. Figure 1 is not properly referred. Line 57 cites Fig. 1d, but there are no mentions to panels a-c before.
6. In line 81, it should read Supplementary figure because it is the beginning of phrase.